# Integrated mechanical computing for autonomous soft machines

Junghwan Byun[1,2,7], Aniket Pal [1,3,7], Jongkuk Ko[1,4] & Metin Sitti [1,5,6] ✉

Mechanical computing offers a new modality to formulate computational autonomy in intelligent matter or machines without any external powering or active elements. Transition (or solitary) waves, induced by nonreciprocity in mechanical metamaterials comprising a chain of bistable elements, have proven to be a key ingredient for dissipation-free transmission and computation of mechanical information. However, advanced processing of mechanical information in existing designs is hindered by its dissipation when interacting with networked logic gates. Here, we present a metamaterial design strategy that allows non-dispersive mechanical solitary waves to compute multi-level cascaded logic functions, termed 'integrated mechanical computing', by propagating through a network of structurally heterogeneous computing units. From a perspective of characteristic potential energy, we establish an analytical framework that helps in understanding the solitary wave-based mechanical computation, and governs the mechanical design of key determinants for realizing cascaded logic computation, such as soliton profile and logic elements. The developed integrated mechanical computing systems are shown to receive, transmit and compute mechanical information to actuate intelligent soft machine prototypes in a seamless and integrated manner. These findings would pave the way for future intelligent robots and machines that perform computational operations between various non-electrical environmental inputs.

Intelligence embodied in biological cellular architectures often appears to mediate macroscopic dynamic responses to external mechanical stimuli. For example, mechanotransduction of sensitive plants, such as the Venus flytrap (*Aldrovanda vesiculosa*) or *Mimosa pudica L*[1,2]., leads to controlled opening and closing of their leaves, as the mechanical outcome of biochemical operations in response to environmental mechanical stimuli. Although seemingly binary, this behavior requires intelligent "computation" involving responsiveness to time-stamped inputs (e.g., two consecutive mechanical stimulations of the same sensory hair)[3–5] or sensitivity to intensity[6,7], implying that computational operations between non-electrical signals engage deeply in the formulation of intelligent "living machines".

Similarly, the design of computational responses in synthetic architected materials, termed "mechanical computing", is of growing interest in intelligent matter and machines that perform tasks between non-electrical environmental signals, such as soft robots[8]. In this framework, computational operations are executed solely by deformation and transition dynamics of architected materials, thus featuring unconventional, electronics-free information processing with environmental signals being seamlessly coupled. To harness this advantage,

[1]Physical Intelligence Department, Max Planck Institute for Intelligent Systems, 70569 Stuttgart, Germany. [2]Soft Hybrid Materials Research Center, Korea Institute of Science and Technology, 02792 Seoul, Republic of Korea. [3]Institute of Applied Mechanics, University of Stuttgart, 70569 Stuttgart, Germany. [4]Department of Chemical and Biological Engineering, Gachon University, Gyeonggi-do 13120, Republic of Korea. [5]Institute for Biomedical Engineering, ETH Zürich, 8092 Zürich, Switzerland. [6]School of Medicine and College of Engineering, Koç University, 34450 Istanbul, Turkey. [7]These authors contributed equally: Junghwan Byun, Aniket Pal. ✉e-mail: sitti@is.mpg.de

ongoing efforts have been directed to supplant conventional components (e.g., gears and linkages)[9] with soft architected materials entailing mechanical information[8]. Specifically, multistable configurations of mechanical elements that leverage snap-through instabilities of soft elastic beams[10–17], shell[18–21], or origami linkages[22–24] abstract digitization of mechanical information. When triggered, transition behaviors between buckling modes of structurally coupled mechanical bits lead to a cascade of storage and release of elastic potential energy, which in turn results in localization[16,25], propagation in the form of mechanical transition waves[11,13,26], and/or logic operations[11,15,27,28]. Efforts are now growing to employ multiple stimuli and controlling signals (e.g., electronic signals[29–32], magnetic fields[33–36], chemical stimuli[12,37], humidity gradients[22], fluid pressure[18,19,38,39], etc.) that provide opportunities for further advancing the embedded processing capability, programmability, and robotic applications.

Computational autonomy as a material property per se—that is, information can be transmitted, processed, and retained solely through soft material configurations in the absence of external power sources— is the primary advantage of mechanical computing[8,15,28]. As mechanical mechanisms encoded in material configurations account for a larger proportion in information processing, a more advanced form of completely mechanical computation (i.e., autonomous operation without any external powering or active elements) becomes critical. However, the exclusion of active elements in intelligent matter design has largely been overlooked. Indeed, demonstrations of purely mechanical computing processes have so far been limited to single logic gates and the corresponding fundamental logical functions[12,15,22,27,28]. Although it is quite well established by conventional integrated circuits (ICs) that the computation capability is generally proportionate to the number of computing units, currently, no studies have provided meaningful insight to the design solution for computational autonomy in soft matter consisting of multiple mechanical computing units (termed "mechanologics") intertwined with each other[22]. Central to this challenge is the dissipation-free transmission of computational mechanical information through a network of mechanologic clusters without the aid of external intervention or powering. A systematic understanding of mechanical signals that propagate through structurally heterogeneous mechanologics is lacking, despite both propagation in homogeneous media[11,35,40] and operation within stand-alone mechanologics[12,22,27,28] being relatively well studied. Accumulation of potential energy barrier (or mechanical impedance) from cascaded mechanologics acting against the propagating signals also remains an intractable problem.

Here, we present a design framework for computational mechanical metamaterials that enable multi-level cascaded logic computation of propagating mechanical signals. Taking inspiration from a mature design paradigm of electronic system architectures (Supplementary Fig. 1), we demonstrate how the systematic assembly of soft bistable unit elements leads to a monolithic, integrated computing architecture consisting of a network of mechanologics connected via transmission lines (Fig. 1a). In this architecture, digital information is transmitted in the form of non-dispersive mechanical solitary waves, and its computation is performed by the dynamic mechanical coupling of the solitary waves with structural heterogeneity in mechanical computing units. Two characteristic energy terms, the input energy barrier ($U_b$) and the output transmissive energy ($U_{out}$), are introduced to develop a simplified model of solitary wave-based mechanical computation (Fig. 1b). Based on the model, we show that the soliton width, $W_{soliton}$, is the critical factor governing the behavior of both signal transmission and computation, and explore how to design transmission lines and mechanologics through which mechanical information performs dissipation-free cascaded computing without the use of any active elements (Fig. 1c, d). These findings allow us to generalize a design rule for implementing system-level integrated mechanical computing. Proof-of-concept demonstrations show how the developed mechanical computing system can direct mechanical instructions across environmental inputs, computational operations and actuator modules in a monolithic form.

## Results

### Design and characterization of mechanical transmission lines

Key ingredients to build integrated computing architectures generally represent mechanical transmission lines and logic devices, termed "mechanologics" (Fig. 1a and Supplementary Fig. 1). On top of the classical bistable beam theory[41–43] and the mechanical solitary wave theory[11,35], first, we developed a design and characterization strategy of the mechanical transmission line to better understand its quantitative characteristics in a methodical way (Fig. 2). As a unit constituent, a soft bistable element with non-reciprocity was defined by two symmetric tilted beams that are connected at the center, and its possible design space was investigated by both numerical and experimental parametric studies about length ($L$), beam thickness ($t$), and angle ($\theta$) (Supplementary Fig. 2). Note that unlike most previous studies using tilted beams[10,11,13], in our approach, the lattice constant ($a$) should additionally be considered to properly engineer the bistable characteristics[44] (Supplementary Figs. 2 and 3). Given the experimental feasibility and geometrical stability, the optimized beam geometry was chosen to have a parameter set of ($a$, $L$, $t$, $\theta$) = (9 mm, 12 mm, 0.6 mm, 25°) featuring a net transmissive energy $\Delta U^1$ ($=U_{out}^1-U_b^1$) of -0.15 mJ (Fig. 1b, Supplementary Fig. 2 and Methods for optimal design consideration). A 1-dimensional (1D) series connection of the optimized bistable elements via spring-like linear coupling units constructed a mechanical transmission line which can produce a non-dispersive mechanical solitary wave by a chain of non-reciprocal snap-through dynamics[11] (Fig. 2a; see also Fig. 1c-(i), Supplementary Fig. 4 and Supplementary Movie 1). To quantify the nature of this solitary wave propagation, we established a simplified model of the transmission line by introducing two characteristic energy components, $U_b$ and $U_{out}$ (Fig. 2b; see also Fig. 1c-(i) and Methods). Specifically, given the transmission line consisting of $N$ ($=N_1+N_2$) bistable elements, we reasoned that both the input energy barrier, $U_b^{T(N)}$, and the transmissive energy delivered to the output end of the transmission line, $U_{out}^{T(N)}$, are heavily reliant on the soliton width ($W_{soliton}$)[45] or equivalently the parameter $N_{soliton}$ ($\approx W_{soliton}/a$), the number of springs constituting a soliton at any given time frame. This implies that $U_b^{T(N)}$ and $U_{out}^{T(N)}$ can be approximated respectively as $U_b^{T(N_{soliton})}$ and $U_{out}^{T(N_{soliton})}$, suggesting the sufficient condition for the dissipation-free transmission to be $U_{out}^{T(N_{soliton})} > U_b^{T(N_{soliton})}$ (Fig. 2b; see Methods for the details). The experimental characterization of transmission lines with varying $N$ reveals that, with the increase of $N$, both $U_b^{T(N)}$ and $U_{out}^{T(N)}$ asymptotically reach steady-state values, validating the existence of $N_{soliton}$ (Fig. 2c, d and Supplementary Fig. 5). We also see that the $N_{soliton}$ value and the absolute potential energies significantly change as a function of spring stiffness, $k$ (Fig. 2c, d and Supplementary Fig. 5). This trend is in good agreement with numerical results showing that the $N_{soliton}$ value can be engineered from 2 to 5 in average (temporarily up to 7) as a function of the normalized spring stiffness ($k/k_0$, with $k_0$ equal to 180 N m⁻¹) (Fig. 2e and Supplementary Fig. 6). With the fixed beam geometry; therefore, the spring stiffness ($k$) becomes the only decisive factor governing the behavior of the transition wave-based signal transmission[25] (see Supplementary Fig. S7 for the effects of $k$ and linkage geometry on solitary wave motion). Based on the above investigations, small $k$ (=180 N m⁻¹) was adopted for our transmission line design, which provides not only a relatively low energy barrier (≈0.47 mJ) for propagation, but also small $N_{soliton}$ (=2-3) for compact system design. The resultant specification of the transmission line

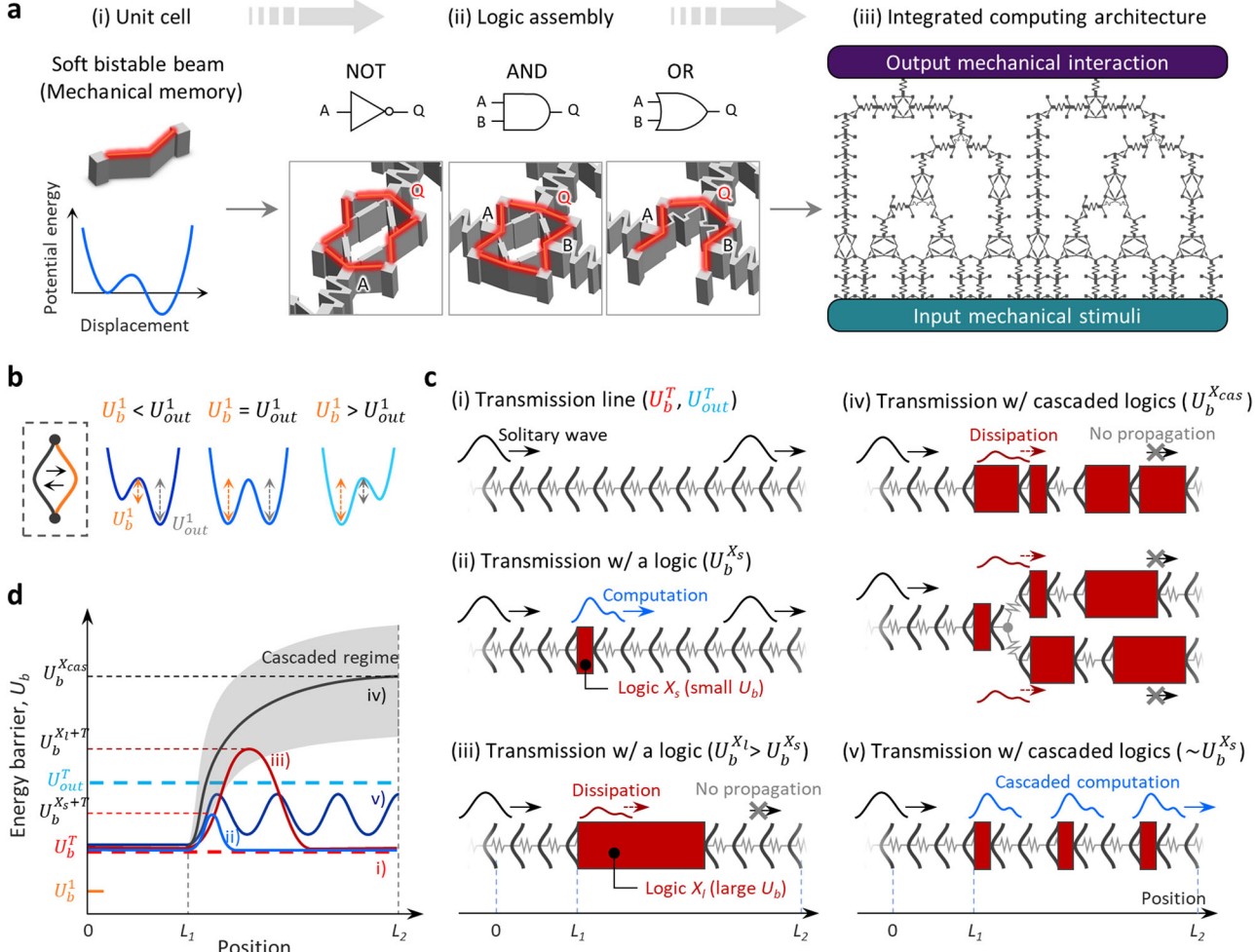

**Fig. 1 | Design concept of integrated mechanical computing. a** Unit cells, their topological assembly for computational logics, and system architectures for integrated mechanical computing. **b** Schematic of a unit bistable element with programmable characteristic energy components: the input energy barrier, $U_b^1$, and the output transmissive energy, $U_{out}^1$. **c** A set of schematics describing solitary wave-based mechanical computing processes: solitary information that propagates through mechanical transmission lines with (i) the input energy barrier, $U_b^T$, and the output transmissive energy, $U_{out}^T$, satisfying $U_b^T < U_{out}^T$, (ii) a mechanologic having

small $U_b^{X_s}$ such that $U_{out}^T > U_b^{X_s+T}$, which permits computational propagation, (iii) a mechanologic having large $U_b^{X_l}$ such that $U_{out}^T < U_b^{X_l+T}$, which leads to the transmission failure, that is, information dissipation, (iv) a cascade of mechanologics characterized by accumulated mechanical impedance that increases the input energy barrier for computation, and (v) our design approach featuring a cascade of rationally designed mechanologics to alleviate the coupled $U_b$ issue. **d** Qualitative $U_b$ profiles as a function of position for the solitary wave propagations shown in **c**.

features $N_{soliton}$ of 2-3, $U_b^T$ of -0.47 mJ and $U_{out}^T$ of -0.73 mJ, satisfying the sufficient condition for propagation $U_{out}^T > U_b^T$.

## Computational propagation through a structurally heterogeneous computing unit

A transition wave-based computing event, namely "computational propagation", occurs when the propagating soliton meets the structural heterogeneity of logic elements (Fig. 2f and Supplementary Movie 1). The presence of such heterogeneity locally increases the effective mechanical impedance ($R$) compared to that of the transmission line unit ($R_0$) (Supplementary Fig. 8), and accordingly induces transient changes in soliton profile as a consequence of the computing process (Fig. 1c). Note that $R$ is closely correlated with $U_b$, but represent the effect of each segment unit more intuitively. To systematically investigate this computational propagation, we modified the model shown in Fig. 2b by inserting a mechanologic component $X$ (Fig. 2g), and carried out numerical simulations in which the logic $X$ was abstracted as a spring with variable mechanical impedances

($R > R_0$) (Supplementary Fig. 8 and Supplementary Movie 2). Figure 2h–m shows typical numerical results involving the evolution of the normalized displacement for each bistable unit in computational propagations with $R$ ranging from $R_0$ to $20R_0$. Herein, the normalized displacement is defined by the net displacement of each bistable beam (i.e., $|x_i - x_{i0}|$, where $x_i$ is the position of the $i^{th}$ bistable element with its initial position $x_{i0}$) in the propagation direction divided by the average maximum displacement of all the actuated beams ($d_{max}$). To visualize the shape and time-dependent propagation of solitary signals, we used the instability factor ($S_i$)[11], which implies the beam's normalized distance from the nearest stable configuration, defined by $\min(|(x_i - x_{i0})/d_{max}|, |(x_{i0}+d_{max} - x_i)/d_{max}|)$. Notably, the obtained $S_i$ profiles suggest that although time-dependent $S_i$ was altered with respect to the homogeneous case (Fig. 2h, i), the soliton width ($W_{soliton}$ or $N_{soliton}$) remained nearly unchanged for all computing events in every time domain of interest (Fig. 2j–m and Supplementary Fig. 8l). This result verifies that $W_{soliton}$ prescribes the physical boundary of solitary wave-based mechanical computation as well as transmission. Our finding further suggests that if

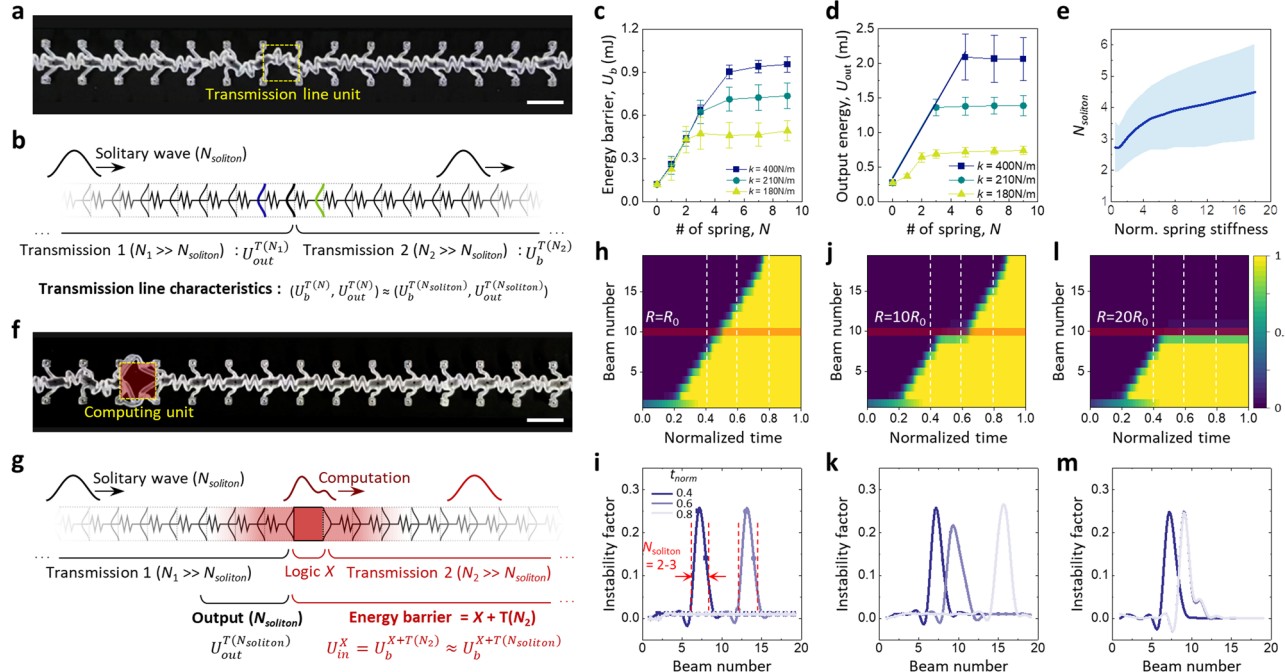

**Fig. 2 | Design and characterization of solitary wave-based computational propagation. a** Typical propagation behavior of mechanical solitary information through a mechanical transmission line (Supplementary Movie 1). Scale bar, 1 cm. **b** Schematic diagram of the mechanical transmission line consisting of $N$ ($= N_1 + N_2$) elements with input, $U_b^T$, and output, $U_{out}^T$ characteristic energy components (see Methods). **c, d** Experimentally obtained $U_b^T$ and $U_{out}^T$ of transmission lines with $N$ varying from 0 to 9. **e** Numerically estimated $N_{soliton}$ as a function of the normalized spring stiffness ($k/k_0$) with $k_0 = 180$ N m$^{-1}$. **f** Typical propagation behavior of mechanical solitary information through a transmission line that contains a structurally heterogeneous computing unit (Supplementary Movie 1). Scale bar, 1 cm.

**g,** Schematic diagram of the transmission line connected with a mechanical computing unit $X$ (see Methods). **h, j, l** Numerical studies on the evolution of the normalized displacement of each bistable unit in homogeneous transmission (mechanical impedance, $R_0$) (**h**) and computational transmission through a computing unit having a mechanical impedance ($R$) of $10R_0$ (**j**) and $20R_0$ (**l**) (Supplementary Movie 2). The variable mechanical impedance is defined only for the tenth spring. A color bar indicates normalized displacement. **i, k, m** The time-evolving propagation behavior of the instability factor ($S_i$) along with the system shown in **h, j, l**, respectively. All error bounds indicate SD.

the length of the mechanologic $X$ is sufficiently shorter than $W_{soliton}$, the sufficient condition for successful computational propagation can be simplified as $U_{out}^{T(N_{soliton})} > U_b^{X+T(N_2)} \approx U_b^{X+T(N_{soliton})}$ (Fig. 2g and Supplementary Fig. 9).

## Design and characterization of mechanologics

To meet the above criterion for dissipation-free computational propagation, we devised a rational design strategy for mechanologics. Differentiated from other previous studies[11,12,15,27,28], we focused on developing the most compact design of force transmission by which at most four bistable beams achieve various sequences of buckling mode transitions within a unit square lattice (length, $a$) (Fig. 3a). Based on this backbone structure, subtle tuning of each of their potential energy profiles led to a complete set of logic functions (NOT, AND, and OR) within the unit dimension that exactly matches the transmission line unit (Fig. 3a, b, Supplementary Figs. 10 and 11, and Methods). The benefit of this mechanologic design allows for (i) superior system compactness, (ii) size matching and seamless connection with transmission lines, (iii) systematic analysis of computational propagations, and (iv) simplification of energy barrier design for solitary wave-based mechanical computing (Supplementary Fig. 9). In the resulting design, the mechanical solitary wave transmitted through the transmission line is seamlessly coupled with the mechanologics' beams, and subsequently creates computational functions as a consequence of combinatorial buckling behaviors (Fig. 3b and Supplementary Movies 3–5). The feasibility of computational propagation was experimentally analyzed by the input/output characteristics of the mechanologics with transmission lines ($N_2 = 6$) connected after them (Fig. 3c). All logic

devices were carefully designed to have bistable characteristics with input/output potential energies satisfying the criterion for propagation: their energy barriers were engineered to be lower than $U_{out}^T$ ($\approx 0.73$ mJ), thereby permitting successful computational propagation without loss of information (Fig. 3d). Importantly, we note that the experimentally obtained values of energy barriers ($U_b^{X+T(N_2)}$) start to saturate at the regime of $N_2 = N_{soliton}$ ($= 2–3$) (Fig. 3d), which perfectly matches our finding that $N_{soliton}$ is a key decisive factor for transition wave-based computational propagations. The operating performance of the proposed mechanologics shows explicit computational propagations which were quantitatively analyzed by the evolution of normalized displacement of each bistable beam (Fig. 3e, Supplementary Fig. 12 and Supplementary Movies 3–5).

## Computational propagation through multiple cascaded mechanologics

Achieving integrated computing is equivalent to addressing the effective impedance accumulated from multiple cascaded logic devices (Fig. 1c, d). In general, the first energy barrier of the 1D cascaded computational system ($U_b^{1st}$) can be expressed by (Fig. 4a):

$$U_b^{1st} = U_b^{\sum\{X_i + T(L_i)\}} \approx U_b^{\sum\{X_i + T(N_{cas,i})\}} \quad (1)$$

where $X_i$ is the $i^{th}$ mechanologic, $L_i$ the length of the transmission line that bridges the adjacent computing units $X_i$ and $X_{i+1}$, and $N_{cas,i}$ the number of intermediate cascaded springs, calculated as $[L_i/a]$ with the bracket indicating the rounding function to an integer. Despite this cumulative form, our finding on the rule for computational

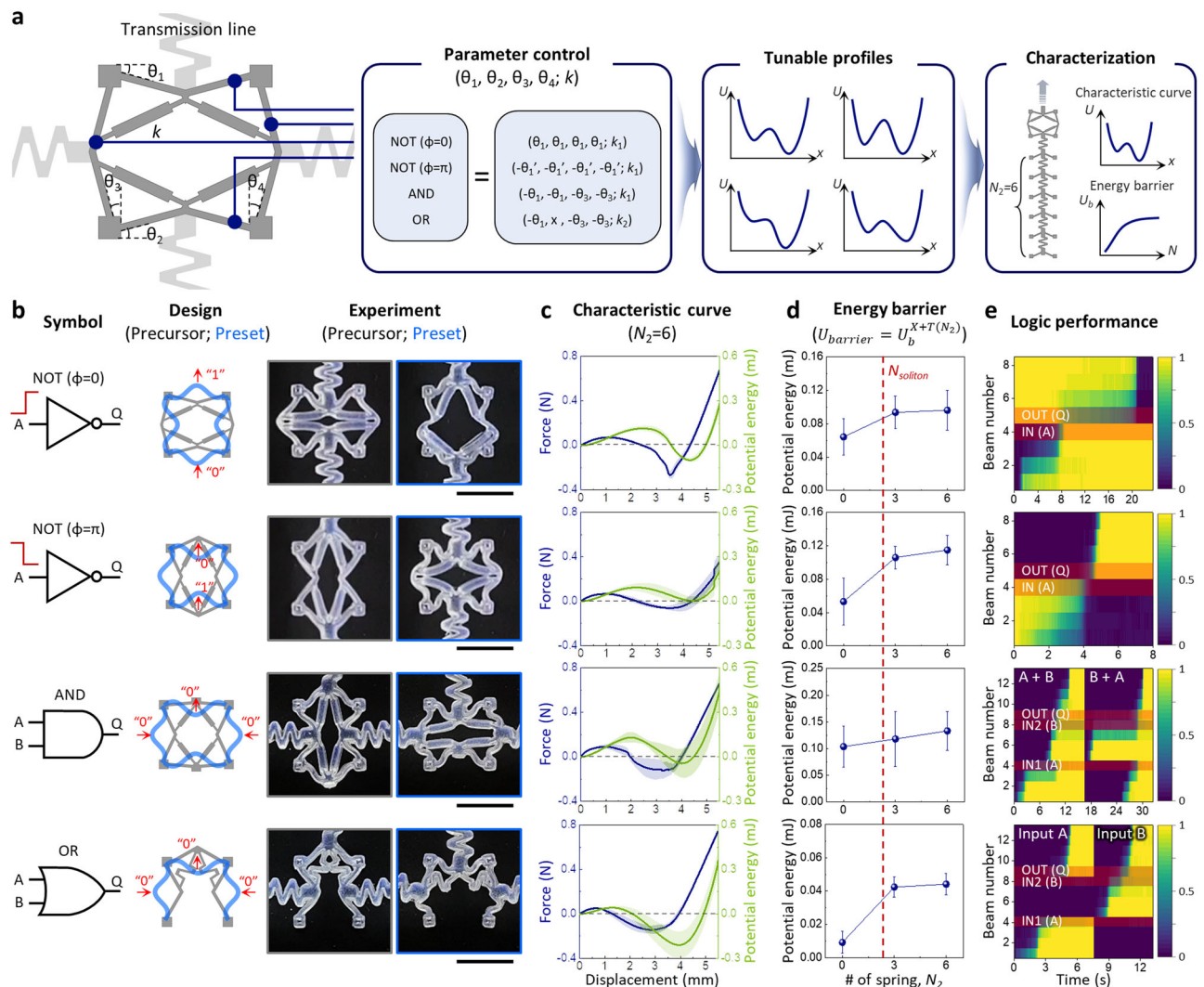

**Fig. 3 | Architected design and characterization of mechanologics. a** Design strategy for mechanologics defined within a unit square lattice. **b** Schematic symbols, architected designs, and photographs of computational mechanologics (NOT, AND, and OR gates). The as-fabricated precursor designs, highlighted in gray, are transformed into the preset buckled states, highlighted in blue, when placed into the lattice frame. See Supplementary Fig. 10 and Methods for the details on the mechanologic design. Scale bar, 1 cm. **c** Experimental characterization of bistable profiles of the mechanologics with transmission lines ($N_2 = 6$) connected behind

them. **d** Experimental characterization of effective energy barriers against computational propagation through the mechanologics as a function of the length (characterized by $N_2$) of the transmission line connected behind it. **e** Time-evolving normalized displacement data showing the experimental performance of mechanologics (Supplementary Movies 3–5). Detailed beam configurations for the analyses are shown in Supplementary Fig. 10. Color bars indicate normalized displacement. All error bounds indicate SD.

propagation, as shown in Fig. 2g, suggests that such coupled energy barrier problems can be decoupled into individual computing steps only if the dimension of mechanologics is smaller than the soliton width ($\approx aN_{soliton}$) and $N_{cas} \geq N_{soliton}$ holds, which reads (see also Fig. 1c, d):

$$U_b^{k^{th}} = U_b^{\sum_{i=k}\left\{X_i + T(N_{cas,i})\right\}} \approx U_b^{X_k + T(N_{soliton})}. \qquad (2)$$

To quantitatively validate this approach, we conducted a set of numerical studies investigating the behavior of computational propagation through multiple cascaded mechanologics with various effective impedances and $N_{cas}$ (Fig. 4b). The results suggest that while cascaded systems with small $N_{cas}$ ($<N_{soliton}$) failed in transmission as well as computation through the logics with relatively large $R$ due to their coupled energy barrier effects (Fig. 4c-(i, ii), and Supplementary Movie 6), the increase in $N_{cas}$ successfully settled down the barrier,

enabling the signal to smoothly pass through the logics with larger $R$ (Fig. 4c-(iii, iv), and Supplementary Movie 6). Experimental verification of this numerical prediction was carried out by implementing 1D chains of mechanical inverters (NOT gates with different phases, $\phi = 0$, $\pi$) with $N_{cas}$ varying from 0 to 3 (Fig. 4d and Supplementary Fig. 13). As a reference, we first analyzed the propagation behavior of the bare transmission line (Fig. 4d-(i) and Supplementary Fig. 13a) and one with a single inverter (Fig. 4d-(ii) and Supplementary Fig. 13b). The primary area of focus was the time taken for propagation of each beam, which approximately alludes to the effective impedance. The results showing the nearly uniform time duration of propagation (Fig. 4d-(i)) and its gradual relaxation through the transmission line connected behind the inverter (Fig. 4d-(ii)) are in very good agreement with the behavior expected in Fig. 1d. Further studies of additional series connection of inverters reveal that in cascaded systems with $N_{cas}<N_{soliton}$, the computational propagation was inhibited even at the initial stage (Fig. d-(iii)) or severely delayed (Fig. 4d-(iv)) due to the cumulative impedance effect (Supplementary Movie 7). On the contrary,

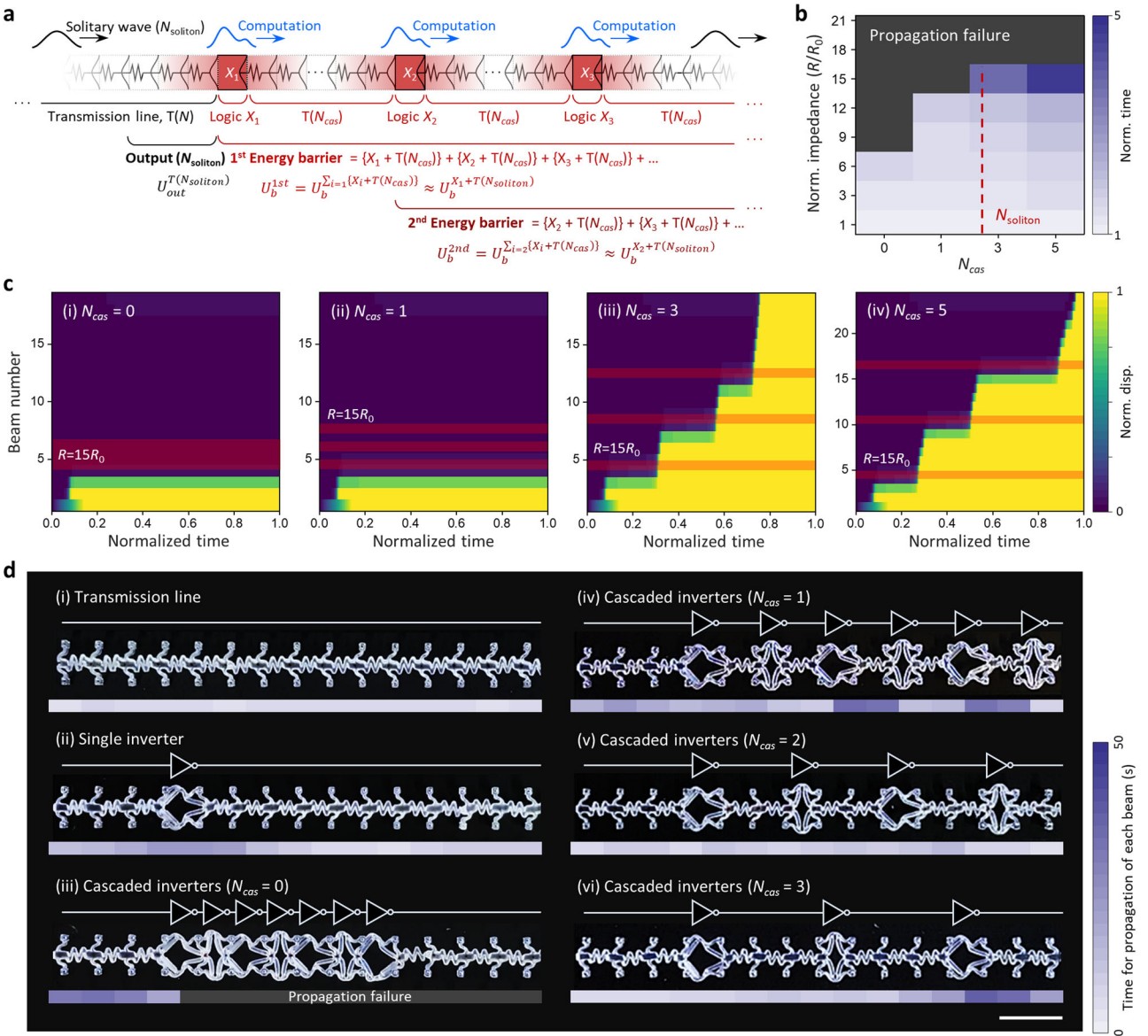

**Fig. 4 | Computational propagation through one-dimensionally (1D) cascaded mechanologics. a** Schematic description of the general design rule for cascaded computation. **b** Simulation results of cascaded computations through a 1D network of mechanologics (mechanical impedance, $R$) connected by transmission lines (mechanical impedance, $R_0$) with varying $N_{cas}$ (0 to 5). The normalized time taken to complete the propagation was introduced not only to characterize the propagation performance, but to indirectly estimate the effective mechanical impedance of overall cascaded systems. **c** Numerical studies on the evolution of the normalized displacement of each bistable unit in computational propagations through cascaded mechanologics (mechanical impedance, $R = 15R_0$) with $N_{cas} = 0$ (i), 1 (ii), 3 (iii), 5 (iv) (Supplementary Movie 6). **d** Experimental verification of the proposed design rule for cascaded computing (Supplementary Movie 7). To demonstrate 1D cascaded computing, mechanical inverters with two different phases ($\phi = 0, \pi$) were connected in series. The time parameter, which has a linear relationship with mechanical impedance for propagation, was obtained for every bistable beam to characterize the behavior of cascaded computing. Scale bar, 2 cm.

the increase in $N_{cas}$ ($>N_{soliton}$) allowed for the repetitive settling down of energy barriers as described in Fig. 1d, thereby achieving successful cascaded computation without conspicuous propagation delay (Fig. 4d-(v,vi) and Supplementary Movie 7).

We equally extended this design principle to the 2D cascaded mechanical computing problem. To fulfill the functional completeness and plausibility of system hierarchy, we designed mechanical logic circuits with 3-level hierarchy consisting of AND, OR and NOT gates, with the output $Q$ defined by $Q = \overline{A_1B_1 + A_2B_2}$ (Fig. 5a). Importantly, distinct from the electrical circuit, the unit components for signal redirection or bifurcation should be also considered to have effective mechanical impedances larger than $R_0$, due to their structural heterogeneity (Supplementary Fig. 14). Computational propagations through two different mechanical circuit layouts ($N_{cas} = 0$ and 2)

suggest that the cascaded design with $N_{cas} = 2$ exhibited successful integrated computing (Fig. 5d, e and Supplementary Movie 8), whereas the computing process with $N_{cas} = 0$ was totally inhibited by the substantial energy barrier that was gradually accumulated from cascaded mechanologics and redirector units (Fig. 5b, c and Supplementary Movie 8).

## Autonomous soft machines based on integrated mechanical computing

The proposed design principle for integrated mechanical computing provides an effective computing paradigm that physically instantiates intricate computational abstractions between mechanical inputs and outputs. To show potential soft machines interfaced by such a computational framework, we present a proof-of-principle demonstration

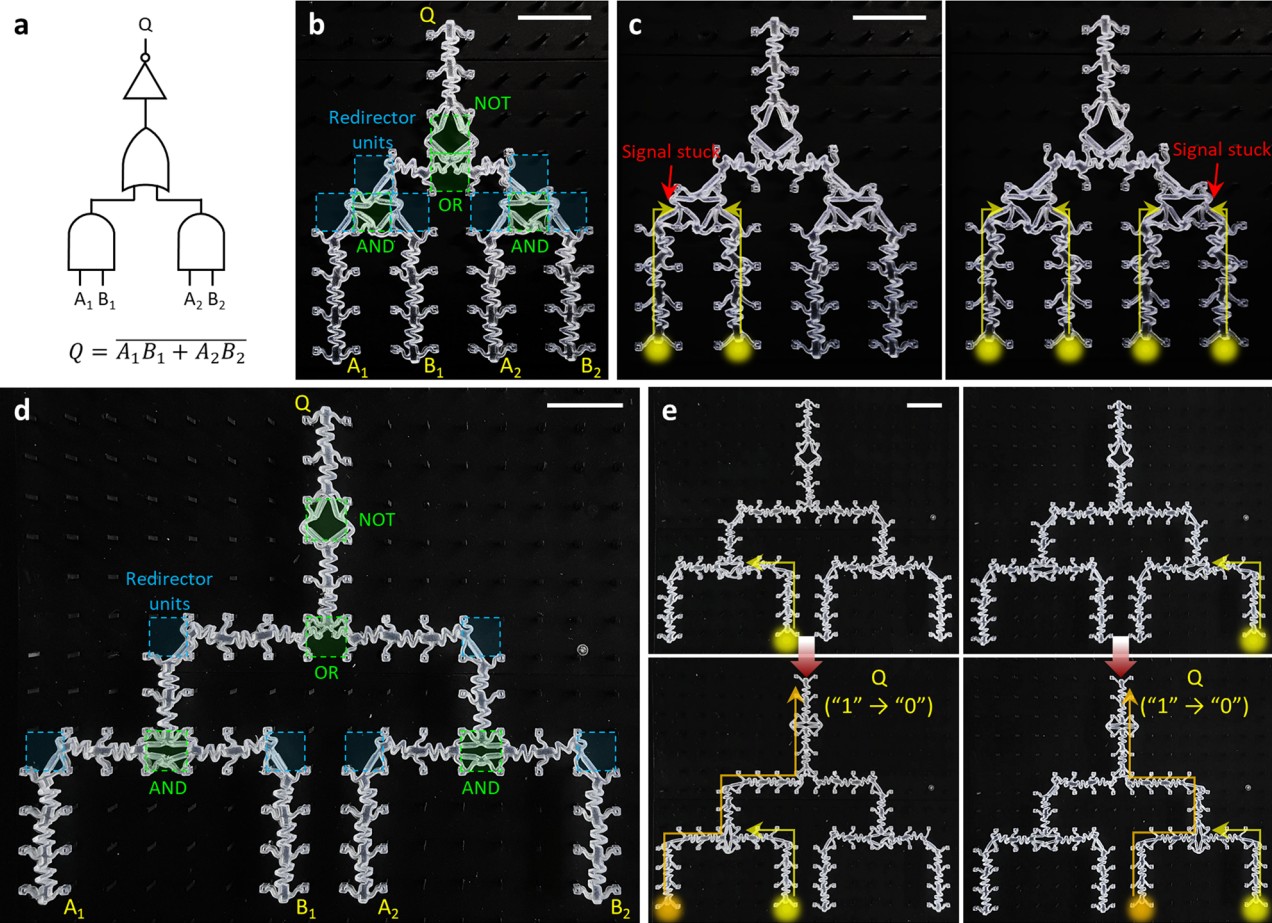

**Fig. 5 | Computational propagation through two-dimensionally (2D) cascaded mechanologics. a** A logic diagram example for the 2D cascaded computing demonstration. **b,d** Physical implementation of the 2D integrated mechanical computing circuit shown in (**a**). The system consists of four mechanologics and six redirector units, networked by transmission lines with $N_{cas} = 0$ (**b**) and $N_{cas} = 2$ (**d**). **c, e** Operation of the mechanical circuit upon stimulation of mechanical inputs ($A_1$, $B_1$) (left) and ($A_1$, $B_1$, $A_2$, $B_2$) (right) (Supplementary Movie 8). The computing failure for **c**, highlighted in red, was due to the excessive energy barrier accumulated from cascaded mechanical components. By contrast, the increase in $N_{cas}$ permitted successful propagation through 3-level cascaded mechanologics (**e**). All scale bars, 2 cm.

of Mimosa plant (*M. pudica*)-inspired autonomous, intelligent, integrative soft machines (Fig. 6). A close observation of *M. pudica*'s thigmonastic movements shows that the plant's response to the mechanical input is differentiated depending on the intensity of the stimulus[6,7,46,47] (Fig. 6a and Supplementary Movie 9). We reinterpreted this stimuli-responsive operating mechanism by means of integrated mechanical computing to realize electronics-free intelligent soft machines (Supplementary Fig. 15).

First, we developed a soft actuator consisting of a 3D printed base layer and an active hydrogel layer, with a design that maximized the actuation rate and strain upon water absorption (Fig. 6b and Methods). The monolithically fabricated reservoir could transport a water droplet (~30 µL) to the actuator (Fig. 6c and Supplementary Fig. 16), which in turn leads to *M. pudica*-like shape morphing with a bending angle of up to 50° (Fig. 6d, Supplementary Fig. 17 and Supplementary Movie 10). A series connection of ten soft actuators via a transmission line demonstrated a soft machine, where a single mechanical input triggered the sequential propagation of actuation events, reminiscent of one branch of *M. pudica* (Fig. 6e and Supplementary Movie 11). Additionally, we implemented an integrated mechanical computing circuit whose logic function was based on the truth table that maps mechanical outputs to three inputs (S1, S2, S3) with different energy barriers (Supplementary Fig. 18a–d).

The developed integrative machine consists of 17 soft actuators (arranged within two different branches) and 104 bistable elements in total whose strategic assembly constructs a computational network between 4 mechanologics and 13 redirector components (Fig. 6f and Supplementary Fig. 18e). This indicates that an initial environmental trigger can demonstrate computational propagations of mechanical information through a network of 17 structurally heterogeneous mechanical units including 4 mechanologics. Note that the whole circuit was designed in a double-layer layout using two sub-level circuits (Supplementary Fig. 18); therefore, mechanical signals can propagate through the two layers independently and without interacting with each other, enabling us to reduce the areal footprint as well as allow for multi-channel signal crossings. With input stimuli S2 and S3, the signal initiates concurrent and independent propagations along two different layers to actuate the two parts of branch 1 simultaneously, whereas for input S1 the signal propagates along only one layer and actuates only one part of branch 1 (Fig. 6g, h and Supplementary Movie 12). This intelligent operation of the machine shows that depending on the input type, proper mechanical instructions could be propagated through and interact seamlessly with computational operations and actuator modules to demonstrate the force-dependent actuation sequences observed in *M. pudica*.

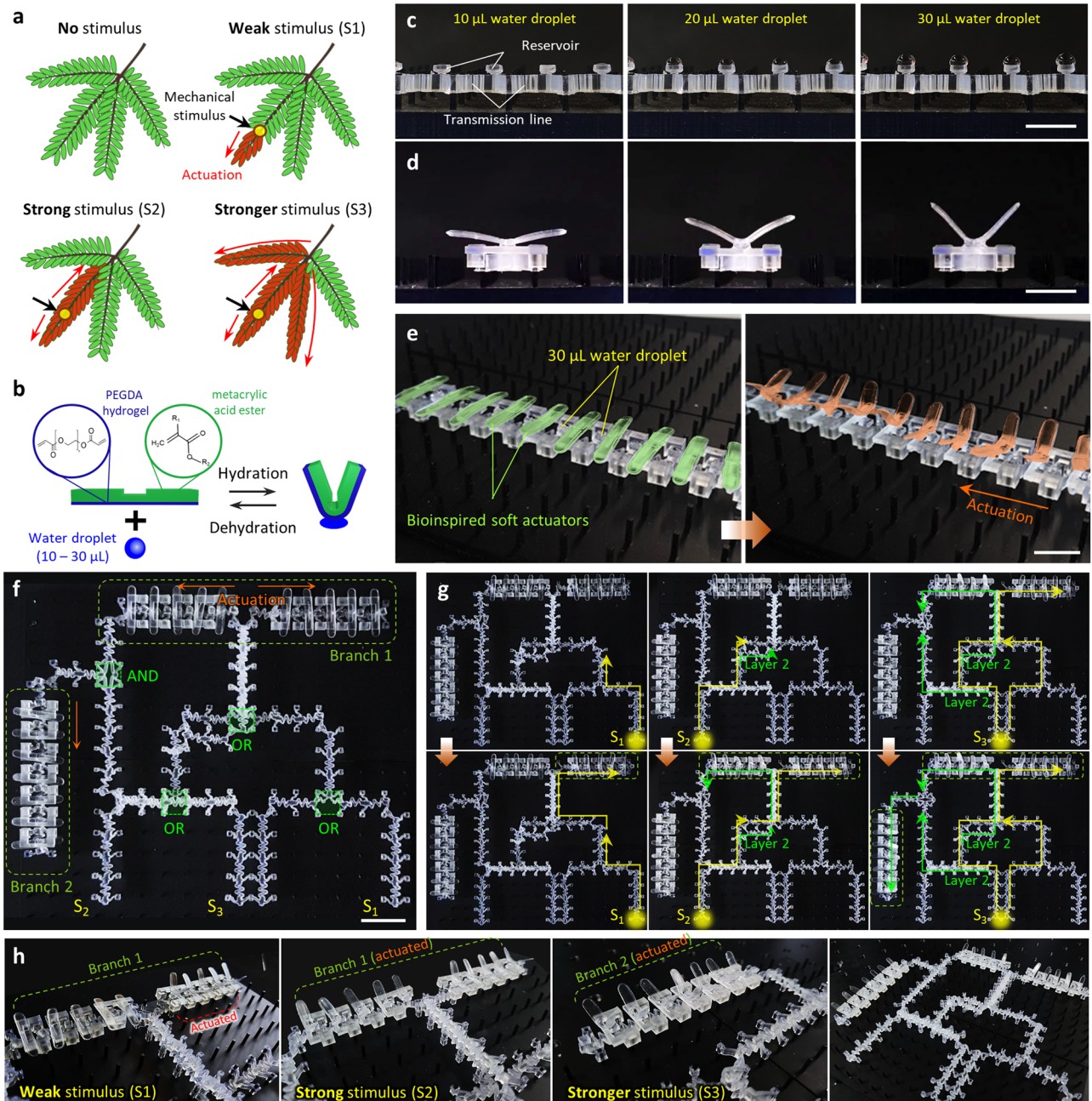

**Fig. 6 | Autonomous soft machines based on integrated mechanical computing. a** Schematic illustration of the force-dependent thigmonastic movements of *M. pudica* (Supplementary Movie 9). **b** Schematic illustration of the design and actuation mechanism of soft hydrogel actuators. **c** Photographs of the reservoirs mounted monolithically on the transmission line for water droplet transport with a volume up to 30 µL. **d** Typical actuation behaviors of the hydrogel actuator upon water uptake with volume of 10, 20, and 30 µL, respectively from left to right (Supplementary Movie 10). **e** A soft machine consisting of ten soft actuators connected in series via the mechanical transmission line (left) and its sequential actuation upon mechanical stimulation (right) (Supplementary Movie 11). **f** Photograph of the autonomous soft machine that reproduces the function of the internal operating circuit of *M. pudica* as described in (**a**). The system consists of double-layered integrated mechanical computing circuits (1 AND gate and 3 OR gates) and 17 soft actuators arranged within two different branches. **g, h** Operation of the integrated mechano-intelligence (**g**) and resulting force-dependent actuation sequences (**h**) (Supplementary Movie 12). Scale bars, 1 cm (**c**–**e**) and 2 cm (**f**).

## Discussion

We have developed a design framework for metamaterial architectures that could execute multi-level cascaded computational functions of mechanical information using purely mechanical computing processes. Our approach of modeling and analyzing computational propagations of solitary waves, together with the compact, rational and seamless design of mechanologics, allowed us to overcome major challenges in cascaded computing where solitary information can be transmitted through a network of mechanologics in a lossless manner.

The outcome was the electronics-free, integrated mechanical computing that realized integrative mechanical-to-mechanical operations directly mapping environmental mechanical stimulus inputs to soft actuators.

The primary advantage of mechanical computation platforms is that they are able to perform logical computations and retain information in the absence of external power sources[8]. This computational autonomy is attributed to completely mechanical computation according to soft material configurations (without any external

powering/active elements); however, existing studies have so far been limited to demonstrations of single logic gates, limiting the number of logical functions to one[12,15,22,27,28]. This limitation stems, as we show, from both fundamental (all information is contained in a single soliton width) and engineering (repeatably fabricating transmission lines with small soliton widths and designing compatible, rationally designed logic elements) requirements which had not been systematically investigated so far. Apart from a solitary wave-based approach, Ion et al. proposed a disconnected spring system containing rigid "signal blockers" for purely mechanical computation[15]. While novel, the system also suffers from the same drawback of being limited to a single logical computation. Additionally, the strategy requires, apart from the "input" signals, a separate evaluation signal which increases complexity and physical footprint while reducing the scope for autonomous actuation based on environmental signal (e.g., an AND gates requires not only both inputs to be 1, but also an additional evaluation signal to change the signal to 1; this would prevent the system to autonomously respond to two environmental inputs and instead have to wait for a manually sent evaluation signal to change the output). As a result of our methodical and parametric exploration, we are able to, for the first time, demonstrate complex yet purely mechanical integrated logic computation with autonomously propagating mechanical information. In the proposed platform, an intrinsic invariance of solitary information to the shape of excitation signals renders the computational capability robust[40]; that is, while the snap-through response of the first element would change with different types of the input signal, the waveform of the soliton is unchanged during its propagation. Furthermore, a systematic approach for reducing the soliton width approximately to two spring elements provides a cornerstone for performing cascaded computation while maintaining a compact system form factor.

While this study focuses on the fundamental design strategy for enhancing computational autonomy in completely mechanical soft-matter systems, real-world applications should entail the capability of automated presets of bistable elements and devices. Although the current non-reciprocal bistable design only permitted spontaneous one-way propagation (Supplementary Fig. 2)[11], the elaborate design of energy landscapes in bistable elements[26] and/or the inclusion of active control modalities, such as pneumatic flows[18,19,38,39], electronics[29–32], (electro-)magnets[33–36] or other environmental inputs[12,37], could provide room for reversal of energetically favorable behaviors. Future work in this direction will include the development of active, heterogeneous building blocks that contain stimuli-responsive (e.g., magnetic fields, water, pH, etc.), shape-changing materials (see Supplementary Fig. 19).

The computing speed and scaling-down issues of the proposed mechanical computing system are another major challenge for practical use. First, the factors governing the computing speed are basically twofold: (i) the speed of the "computing process" itself, and (ii) the transmission speed of mechanical solitary waves through the transmission line. The former factor, which is interpreted as sequential transition behaviors between buckling modes of the bistable elements that comprise mechanologics, is heavily reliant on the bistable properties and the corresponding input/output characteristics of mechanologics; the experimentally obtained values (normalized by the lattice constant, a) were $0.06 - 0.33$ s-1 for the cases of NOT gates (see Fig. 3e). On the other hand, the transmission speed determined by the combination of energy-release behavior of each bistable beam (thus, the bistability profile) and conventional elastic wave velocity ($v = \sqrt{E/\rho} \approx \sqrt{k/\rho}$, where $E$ is the Young's modulus and $\rho$ is the density; the approximation is valid under the linear elastic regime). Assuming the snap-through transformation of bistable elements being a spontaneous motion, the timescale of entire computation processes ($\sim v^{-1} \sim k^{-1/2}$) is governed by elastic wave velocity. This investigation suggests that the speed barrier associated with mechanical transduction between adjacent bistable elements varies by orders of magnitude with respect to $k$ (5–200 mm s-1 for our experiments, data not included)[11]. Therefore, increasing the spring stiffness ($k$) would be an option to improve the computing speed, but a close positive correlation between $k$ and $N_{soliton}$ (Fig. 2e) may pose a trade-off between wave speed and system compactness (i.e., integration density). We chose an optimal value of $k$ of 180 N m-1 which leads to $N_{soliton}$ of ~2 and transmission speed of 8–10 mm s-1 (Supplementary Fig. 6b), fulfilling an acceptable computation speed while maintaining a compact form factor. Similarly, given the normalized snap-through force and potential energy of a unit bistable element being scale-independent, $\sim (t/a)^3$ (see Methods), a scaling-down issue is governed by elastic linkage structures whose spring constant ($k$) is linearly proportional to the length scale. In this regard, selective material assignment (e.g., using stiffer materials for the linkage parts) could be considered as a potential solution for both of these issues of computing dynamics, speed and scaling.

Notwithstanding the above limitations for real-world operations, the proposed computing platform provides ample inspiration for intelligent matter design, and will broaden the available options for computational frameworks to realize autonomous intelligent machines specialized in operations between non-electrical environmental stimuli[48]. In particular, the performance, function and potential applications would be vastly extended when in conjunction with 2D transmission waves[13], architected metamaterial designs[21,31,49–51], stimuli-responsive materials[12,52], and soft robots[53–58]. We anticipate that integrated mechano-intelligence would open new routes for bridging the gap between various dynamic environmental factors and artificial machines.

## Methods
### Material characterization
Dogbone-shaped samples were 3D printed with the soft material that comprises bistable elements (Elastic 50 A, Formlabs), and uniaxial tensile tests were then performed according to ASTM D412-C specifications to measure the linear elastic and viscoelastic property, yield stress, and long-term toughness. We used a universal testing machine (Model 5942, Instron®, Illinois Tool Works Inc.) coupled with a non-contacting video extensometer (Model AVE2, Instron®, Illinois Tool Works Inc.) to perform uniaxial tensile tests at a crosshead loading rate of 25 mm min-1 while using a 50 N load cell to measure the force. The Elastic 50 A material was found to be linearly elastic with an elastic modulus of 2.04 MPa and a Poisson's ratio of 0.37 (Supplementary Fig. 2b). A set of step strain experiments with two different strain levels (20%, 40%) was conducted to obtain the viscoelastic behavior of the Elastic 50 A material by applying a fast, instantaneous displacement (4 mm at 120 mm min-1) for 300 s (Supplementary Fig. 2c, d). We then fitted the Maxwell form of the standard linear solid (SLS) model of linear viscoelasticity to our experimental data; where the instantaneous force ($F(t)$) can be represented as $F(t) = F_0 + F_1 e^{-t/\tau}$, where $F_0$ and $F_1$ are the time independent and time dependent components of the total force, respectively, and $\tau$ is the time constant (Supplementary Fig. 2e). The results suggest that the stress relaxation effect was almost negligible after 3 min, and that our SLS model had a high correlation with the experimental data ($R = 0.981$) and showed a very low $F_1/F_0$ ratio (= 0.08), indicating that the material demonstrates primarily pure elastic behavior. Apart from the viscoelastic behavior, we also noticed that the yield stress of the Elastic 50 A material is ~1 MPa (at the yield strain of 50–60%; see Supplementary Fig. 2b) and the maximum local stress applied to the beam in its preset buckling state is ~0.7 MPa (Supplementary Fig. 2g), which implies the maximum local strain of ~40%. Based on this investigation, we additionally performed a step strain experiment of 40% strain until the sample was fully damaged (Supplementary Fig. 2f). The result shows that the developed soft bistable beam could sustain its preset buckling state up to ~75 min (~4500 s).

## Dynamics and design optimization of a soft bistable unit element

Our design strategy for unit bistable elements aims to achieve the compact and functional assembly of bistable beams with different bistable profiles within a unit lattice, which possibly forms into various mechanical computing components. Previous studies using tilted beams[10–12] or 2D triangular tessellation[13] achieved bistable properties in a relatively simple manner, but the attainable range of their input/output potential energies has been limited solely to the geometries of the unit building blocks. We addressed this challenge by introducing an additional independent parameter, the lattice constant ($a$) (Supplementary Figs. 2j and 3). According to this approach, even a bistable beam of the same length ($L$) can have various potential energy profiles depending on the value of $a$ ($<L$). Interestingly, when being placed into the lattice frame, soft bistable beams with predefined biased beam angles ($\theta$) were buckled into almost the same geometry regardless of $\theta$, thereby enabling easy and symmetric design of mechanical computing units composed of several bistable elements (Supplementary Figs. 2j and 3). For the dynamics and general input/output characteristics, classical beam theories suggest approximate solutions for important parameters in a typical bistable force-displacement profile as shown in Supplementary Fig. 2a, which read[41,42]

$$F_1 \approx c_1 \frac{EIh}{a^3}, -F_2 \approx c_2 \frac{EIh}{a^3}, d_1 \approx c_3 h, d_2 \approx c_4 h - c_5 \frac{t^2}{h} \qquad (3)$$

where $c_1$ to $c_5$ are non-dimensional constants, $E$ is the Young's modulus, $I$ is the moment of inertia, and $h$ is the apex height of the buckled geometry. Importantly, here, $h$ is given as a function of $a$ and $L$, such that $h(a, L)$. Introducing the detailed expression of $I$, we obtain:

$$F_1 \approx \frac{c_1}{12} Ebh(a,L)\left(\frac{t}{a}\right)^3, -F_2 \approx \frac{c_2}{12} Ebh(a,L)\left(\frac{t}{a}\right)^3,$$
$$d_1 \approx c_3 h(a,L), d_2 \approx c_4 h(a,L) - c_5 \frac{t^2}{h(a,L)} \qquad (4)$$

where $b$ is the depth, and $t$ is the thickness of the beam. The approximation of neglecting high modes of buckling simplifies the force-displacement profile into the shape as shown in Supplementary Fig. 2a, which provides the estimation of potential energies:

$$U_1 \approx \frac{c_1 c_3}{24} Ebh^2(a,L)\left(\frac{t}{a}\right)^3$$
$$U_2 \approx \frac{c_2}{24} Ebh(a,L)\left(\frac{t}{a}\right)^3 \left\{(c_4 - c_3)h(a,L) - c_5 \frac{t^2}{h(a,L)}\right\} \qquad (5)$$

Based on this theoretical background, we carried out a parametric study for optimizing the design of a soft bistable unit element. A wide set of numerical studies was performed to provide a possible design space for the parameters, $t/L$ and $\theta$ (Supplementary Fig. 2g–i). To reduce the design space, we determined the value of $a$ equal to 9 mm in order to secure a sufficient space in the unit lattice, and the depth (length in the $z$ direction) of all systems was fixed with 4 mm for ease of manufacturing and placement. Aside from the feasible design space obtained from the numerical studies, practical limitations also occurred due to our manufacturing technology. Given the predefined geometric conditions for $a$ and depth, for example, the beam length $L$ needed to be designed within the range between 10 and 13 mm, otherwise it was unable to place the beam into the lattice frame with given $a = 9$ mm (in case $L < 10$ mm), or the beam underwent severe deflection together with excessive compressive stresses when initially buckled (in case $L > 13$ mm). In a similar context, the thickness $t$ should be designed thinner than 0.7 mm. Moreover, the initial beam angle ($\theta$) over 35° induced undesired instabilities in the state of preset buckling. Given all these considerations, therefore, the experimentally feasible range out of the proposed design space was reduced to 10 mm $\le L \le$ 13 mm, 0.5 mm $\le t \le$ 0.7 mm, and $\theta \le$ 30°, which, in turn, leads to the

design space with $t/(L/2) = 0.09$–$0.12$ and $\theta \le 30°$ (Supplementary Fig. 2h, i). Lastly, we set the optimal parameter set of $L = 12$ mm and $t = 0.6$ mm to possess a sufficient amount of potential energies in buckled states for the design of mechanical transmission lines and computing units and while avoiding excessive compressive stresses. Supplementary Fig. 2k, l shows representative bistable profiles of beams ($L = 12$ mm, $t = 0.6$ mm) with $\theta$ equal to 0°, 15°, and 25°. We found that autonomous propagation starting from the preset buckled state was inhibited in the beam with $\theta$ smaller than 15°. Finally, the bistable beam with a parameter set of ($a$, $L$, $t$, $\theta$) = (9 mm, 12 mm, 0.6 mm, 25°) was set as an optimized design, which delivers a net transmissive energy $\Delta U$ (= $U_{out} - U_{in}$) of ~0.15 mJ.

## Characteristic energy analysis for computational propagation

The input (that is, energy barrier, $U_b^T$) and output ($U_{out}^T$) characteristics of a mechanical transmission line were analyzed in terms of potential energies applied to each unit element, a soft bistable beam with the given optimized geometry of ($a$, $L$, $t$, $\theta$) = (9 mm, 12 mm, 0.6 mm, 25°). Specifically, considering that the transmission line of finite length consisting of $N$ ( = $N_1 + N_2 \gg N_{soliton}$) bistable elements is a series of the transmission line 1 ($N_1$) and line 2 ($N_2$) as shown in Fig. 2b, the input energy transferred to the first element of the transmission line 2 (that is, the $N_1 + 1$th element highlighted in green bold in Fig. 2b), $U_{in}^{N_1+1}$, is given by the output energy of the transmission line 1, $U_{out}^{T(N_1)}$. Similarly, the output load of the $N_1$th beam, $U_l^{N_1}$, is determined by the effective energy barrier (or accumulated impedance) caused by the existence of the transmission line 2, $U_b^{T(N_2)}$. An additional point to consider here is that in homogeneous transmission with $N \gg N_{soliton}$, the input/output characteristics of any arbitrary element must be the same. It indicates that the characteristic potential energies of the $N_1 + 1$th beam should be equal to those of $N_1$th (highlighted in black, Fig. 2b), $N_1 - 1$th (highlighted in blue, Fig. 2b), and eventually one down to the critical $N_c$th beam, which delineates:

$$\left[U_{in}^{N_1+1} = U_{out}^{T(N_1)}\right] = \left[U_{in}^{N_1} = U_{out}^{T(N_1-1)}\right] = \ldots = \left[U_{in}^{N_c+1} = U_{out}^{T(N_c)}\right], \qquad (6)$$

$$\left[U_l^{N_1} = U_b^{T(N_2)}\right] = \left[U_l^{N_1+1} = U_b^{T(N_2-1)}\right] = \ldots = \left[U_l^{N_1+N_2-N_c} = U_b^{T(N_c)}\right]. \qquad (7)$$

Combining the second terms in each parenthesis, we can reduce the equations as (Fig. 2b):

$$U_{out}^{T(N_1)} = U_{out}^{T(N_1-1)} = U_{out}^{T(N_1-2)} = \ldots = U_{out}^{T(N_c)}, \qquad (8)$$

$$U_b^{T(N_2)} = U_b^{T(N_2-1)} = U_b^{T(N_2-2)} = \ldots = U_b^{T(N_c)}. \qquad (9)$$

Since $N_c$ can be approximated by $N_{soliton}$ based on the hypothesis, the input/output characteristic potential energies of the transmission line is finally obtained as ($U_b^T$, $U_{out}^T$) $\approx$ ($U_b^{T(N_{soliton})}$, $U_{out}^{T(N_{soliton})}$). Given this expression, a sufficient condition for autonomous propagation within the homogeneous transmission line is obtained as $U_{out}^{T(N_{soliton})} > U_b^{T(N_{soliton})}$.

The above investigation can be equally extended to the case with the inclusion of a computing unit, mechanologic $X$ (Fig. 2f, g). Any mechanical computing system driven by transition wave-based computational propagations can be readily divided into two parts: (i) the transmission line 1 ($N_1$) before the computing unit (mechanologic $X$), and (ii) a series of the logic $X$ and the transmission line 2 ($N_2$) connected after it. As explored in the transmission line case, the input energy transferred to the logic $X$, $U_{in}^X$, is given by the output energy of the transmission line 1, $U_{out}^{T(N_1)}$, such that $U_{in}^X = U_{out}^{T(N_1)} \approx U_{out}^{T(N_{soliton})}$. Similarly, the output load of the $N_1$th beam, which is shared by the transmission

line 1 and the logic $X$, is determined by the effective energy barrier induced by the logic $X$ and the transmission line 2, $U_b^{X+T(N_2)} \approx U_b^{X+T(N_{soliton})}$. Consequently, a sufficient condition for autonomous computational propagation via the mechanologic $X$ is obtained as $U_{out}^{T(N_{soliton})} > U_b^{X+T(N_{soliton})}$. Note that this condition is valid under our assumption that the mechanologic has the characteristic dimension (lattice constant, $a$) shorter than the soliton width; otherwise, dynamics at the output interface of the logic should be additionally considered (Supplementary Fig. 9b, d).

### Estimation of $N_{soliton}$

The $N_{soliton}$ values in our approach were estimated in two different ways: (i) First, we experimentally measured or numerically obtained the time-evolving normalized displacement data of all bistable elements involved in computational propagation, and then counted the number of moving beams at any arbitrary time frame. The state of moving beams was defined as when the normalized displacement had a value between 0.05 and 0.95. The subdivided steps involve counting the number of intersections between the displacement profile of each beam and arbitrary vertical lines on the time axis (Supplementary Fig. 8j). The obtained $N_{soliton}$ data were plotted along with the normalized time (Supplementary Fig. 8k), which produced the average $N_{soliton}$ of each computing event with calculable error bounds represented by SD (Supplementary Fig. 8l). (ii) Second, we obtained $N_{soliton}$ by experimentally measuring the input/output characteristic potential energies of the transmission line with $N$ varying from 0 to 9 (Fig. 2c, d and Supplementary Fig. 5). As well described in the main text, we found that, with the increase in $N$, the force-displacement and energy-displacement curves asymptotically approach the saturated profiles, supporting the existence of $N_{soliton}$ (Supplementary Fig. 5). These two methods yielded exactly the same result showing the $N_{soliton}$ value of ~2.5.

### Design of mechanologics

All mechanologics take the form of bistable square structures, with designs fit compactly into a unit lattice ($a = 9$ mm) (Fig. 3a,b and Supplementary Fig. 10). We designed their computing functions using at most four bistable building blocks, which were arranged perpendicular to each other via linear elastic linkages. NOT gates, which need to have two different phases ($\phi = 0, \pi$) depending on the phase of the input signals, consist of four identical bistable elements with a parameter set of $(L, t) = (12$ mm, $0.6$ mm$)$ together with $\theta = 15°$ (for $\phi = 0$) or $25°$ (for $\phi = \pi$). Note that, within the same unit lattice, we could easily tune the potential energy profiles developed in each of the four bistable elements by changing $L$, $t$ and $\theta$ (Fig. 3a). Given the practical design space (Supplementary Fig. 2k, l), the values we proposed, especially for $\theta$, indicate the minimum values allowing for autonomous computing, that is, computational propagation without additional environmental or human intervention. All the bistable elements in NOT gates are connected with their neighboring components via a linear elastic linkage. The geometry of the linear linkage is divided into two parts having different width, 0.4 and 1 mm, respectively. The difference in bending stiffness between these two parts, which are ~15.6 times $[=(1/0.4)^3]$, makes the linkages behave like a rigid four-bar linkage so as to enable the inversion of transmitted mechanical bits, while the thinner parts permit the rotation in the proximity of bistable beams (Supplementary Movie 3). AND gates basically take a similar form as in NOT gates. Differences include the presence of two input terminals, which are arranged mirror-symmetrically, and additional geometric engineering of bistable beams to carry different potential energies for the input and output terminals (the output one should have a larger energy barrier; see Supplementary Fig. 10). To achieve this, we designed the beam angles of the input ($\theta_1$) and output ($\theta_2$) terminals to have the values of 20° and 5°, respectively, which delivered successful computational propagation (Supplementary Movie 4). Distinct from other mechanologic designs, OR gates consist of three bistable elements connected via a softer and longer elastic linkage. This open-ended design is helpful to decouple the actuations of two independent input beams. The geometry of the spring-like linkage can be replaced by alternative structures, only if its total length matches the distance between the input and output terminals so that the reversion of the output terminal does not affect the non-triggered input. The beam angle ($\theta$) was designed to have the value of 30° for successful computation (Supplementary Movie 5).

### Fabrication of integrated mechanical computing systems

All bistable beams, transmission lines, and logic units were fabricated monolithically by 3D printing using an elastic material (Elastic 50 A) in a stereolithography printer (Form 3B, Formlabs). All 3D printed parts were cleaned by ultrasonicating (USC 1200 T, VWR) in isopropyl alcohol for 30 min, and subsequently post-processed in an ultraviolet (UV) oven (Form Cure, Formlabs) for 90 min at 60 °C. The lattice frames with the lattice constant ($a$) equal to 9 mm were 3D printed using a rigid material (Clear, Formlabs) and spray painted black to provide contrast for better visualization (Supplementary Fig. 3a).

### Mechanical characterization

The uniaxial tensile testing setup same as that for the material characterization above was used to obtain the force-displacement characteristics of the unit bistable beams, transmission lines, and mechanologics (Fig. 3a, right, and Supplementary Fig. 5a). All the prepared samples were placed in the printed lattice frame, which was affixed to the bottom clip. Specifically, two holes spaced 2.5 mm apart were patterned onto every sample and anchored by two separate cylindrical pins (diameter = 0.4 mm) to allow only linear displacement by preventing undesired rotation of the load point. A displacement-controlled loading was applied by specifying a crosshead loading rate of 4 mm min$^{-1}$ and while using a 5 N load cell to measure the force. For most measurements, at least three different samples were prepared for each design and measured at least three times for each sample. All plots represent the average values with error bounds indicating standard deviation (SD).

### Experimental analysis of computational propagation of mechanical signals

Mechanical computing systems with bistable elements being marked with color dots were placed in the lattice frame. The behavior of computational propagation was recorded and then analyzed by a free video analysis and modeling tool (Tracker 6). The obtained displacement data were analyzed in their normalized forms which were defined by the net displacement of each bistable unit in the propagation direction divided by the average maximum displacement (~4.6 mm) of all the reverted beams.

### Finite element analysis

Finite element (FE) analysis was performed to model the dynamic behavior of individual bistable elements as well as the propagation of transition waves along a lattice of multiple bistable elements connected by elastic links. All FE simulations were performed with a commercial finite element solver (Abaqus/Standard, version 2020, Simulia, Dassault Systèmes). Plane strain conditions were assumed to increase computational efficiency and elements of type CPE8 were used to construct the mesh. All simulations involved three steps: (i) An initial static step to compress the beam so that it matches the lattice length. (ii) A following static step with force-controlled loading to invert (preset) the beam to its second stable state. (iii) A final dynamic implicit step with displacement-controlled loading to revert the beam to its first stable state while recording the reaction force. Using a dynamic analysis enabled us to accurately capture the instabilities and

snap-through behavior of the bistable beams. A displacement-controlled loading was necessary to obtain the complete equilibrium path in the force-displacement plane. For the final dynamic step, the two ends of the beams were fixed with an *encastre* boundary condition and the center of the beam displaced vertically. The energy-displacement data were obtained by integrating the recorded force-displacement data. We ensured quasi-static conditions in the simulations by introducing a small damping factor and by monitoring the kinetic energy of the simulations. Transition wave simulations required the same steps, with all the beams being subjected to steps (i) and (ii), and only the first (end) beam being provided a prescribed displacement in the final step. All other beams are allowed to deform freely as the transition wave propagates through the system and the displacement of the midpoint of each beam is recorded to monitor the wave propagation.

### Fabrication of soft hydrogel actuators

A solution of photoinitiator was prepared by dissolving 150 mg of 2,2-dimethoxy-2-phenylacetophenone (99%, Sigma Aldrich) in 1 mL of dimethyl sulfoxide (Sigma Aldrich). Ethylene glycol dimethacrylate (100 μL) and the photoinitiator solution (150 μL) were added in 2 ml of poly(ethylene glycol) diacrylate (PEGDA, Mn 575, Sigma Aldrich). An 8 μL of the prepared PEGDA solution was dropped onto the surface of 3D printed actuator body made up of the Elastic 50 A material (25 mm × 4 mm) and then covered with a slide glass to adjust its thickness to $25 \pm 5$ μm. The confined solution was polymerized by irradiation of UV light (Bio-Link 365, wavelength of 365 nm, Vilber Lourmat GmbH) for 30 min.

## Data availability

All data supporting the findings of this study are available within the paper and Supplementary Information. All other relevant data are available from the corresponding author upon request.

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

## Acknowledgements

J.B., A.P. and J.K thank the Alexander von Humboldt Foundation for financial support. This work was supported by the National Research Foundation of Korea (NRF) grant funded by the Korean government (MSIT) (No. RS-2023-00211936), the Korea Institute of Science and Technology (KIST) Future Resource Research Program (No. 2E33191), Max Planck Society, and European Research Council (ERC) Advanced Grant SoMMoR project with grant no. 834531.

## Author contributions

J.B. and M.S. conceived and designed the research. J.B. conducted all experiments and performed the collection and analysis of the experimental data. A.P. designed and implemented finite element analysis. J.B. and A.P. analyzed the simulation data. J.K. assisted in the fabrication of the Mimosa-inspired soft machines. M.S. supervised the research. J.B., A.P. and M.S. wrote the paper with input from all authors.

## Funding

## Competing interests

The authors declare no competing interests.
