## [Peer Review File · Nature Communications]

Reviewers' comments:

Reviewer #1 (Remarks to the Author):

The authors report on an approach called integrated mechanical computing. The approach uses transmission lines of bistable units that propagate mechanical waves, specifically solitary transition waves. The authors study the mechanics of the design, for their specific soft material, and its implementation into a variety of logic gate architectures.

The basic idea of the current manuscript is to create scalable computing platforms based on propagation of mechanical bistable transition waves in soft polymers. This basic idea is already described in [Raney et al., Stable propagation of mechanical signals in soft media using stored elastic energy. Proc. Natl. Acad. Sci. U.S.A. 113, 9722-9727 (2016)].

The current manuscript takes this basic idea and creates transmission line segments to assemble logic gates together. The transmission and assembly approaches are described in [Ion et al., Digital Mechanical Metamaterials, Proceedings of the 2017 CHI Conference on Human Factors in Computing Systems (New York, 2017).]

The uniqueness of the current manuscript is the mechanical characterization of their system assemblies and study of time-of-propagation for the bistable solitary transition waves. This does not appear to bear significant scientific merit in light of the earlier publication of the basic idea and assembly approaches that are supposedly new in this manuscript. Consequently, the current manuscript represents an incremental addition to two established and published research ideas.

The authors also suggest that their design philosophy is biologically inspired from plants that exhibit bistable behaviors. This is a tenuous suggestion that is not carried throughout the manuscript. There is nothing particularly biologically inspired about the logic gate assembly or methods thereof.

Overall this manuscript does not appear to have great scientific novelty in light of prior published work.

Reviewer #2 (Remarks to the Author):

The authors propose a systematic design strategy for computational metamaterial platforms capable of system-level integrated mechanical computing in the context of autonomous soft

machines. They show how a systematic 2D assembly of engineered bistable unit elements can create a monolithic computational platform consisting of networked mechanical computing devices and how mechanical

information, in the form of nondispersive solitary transition waves, propagates within this platform to perform digital computational functions such as NOT, AND, OR.

The idea is not new but the specific implementations seem to have some degree of originality. Indeed, the use of bistable elements in programmable metamaterials is not new but its systematic use to create the proposed architectures seems to be new. The authors have put significant efforts into showing various architectures and functions. The paper is well-written and deserves consideration in Nature Communications. However, a major revision is necessary.

1) The authors have to state more precisely what is new in their implementation with respect to the wide literature on mechanical computing.

2) The overall idea of mechanical computing is interesting but the speed barrier associated with mechanical transduction between adjacent bistable elements is not discussed.

3) An important issue is the role played by impulsive or general dynamics excitations. If architected metamaterials made of bistable elements such as those here proposed are subject to dynamic excitations introduced at different points along the chains, the mechanical response can become rather chaotic. What happens in terms of computational capability?

Please improve the general quality of the figures. Sometimes it is really hard to read the labels or the numbers given the high density of subfigures.

Reviewer #3 (Remarks to the Author):

The paper proposes a technique for implementing mechanical computers that take inspiration from biological systems. The paper describes a novel computing cell architecture, analytical model for the activation thresholds and mechanical and numerical tests have been conducted. Finally, it

demonstrates the viability of the approach by replicating the computational characteristics of a Mimosa plant.

The paper is well written and exhibits a high rigor that deepens the understanding of the proposed structures and provides a solid foundation to build on for future work.

The paper gives a broad overview of approaches in other disciplines for implementing mechanical computation. I am missing a short discussion of the delta of this work over e.g. "Digital Mechanical Metamaterials" by Ion et al.. Especially the advantages of propagating signals using the solitary wave approach over disconnected springs should be highlighted. The authors describe a "lack of systematic investigations on cascaded and integrated mechanical computing" in the related work. While there is always room for improvement, it seems to me like an overly generalizing statement. I would prefer, if the authors state which systematic investigations they are contributing instead.

Overall, I would recommend moving forward with this manuscript, as it is a valuable step toward implementing mechanical, biologically inspired computing systems.

Response Letter

We would like to thank the reviewers for their valuable, detailed and constructive feedback, which has helped to strengthen our manuscript. We have addressed all comments and suggestions provided by the reviewers point by point below. The changes made to the manuscript are highlighted in red in the revised manuscript.

Reviewer #1

The authors report on an approach called integrated mechanical computing. The approach uses transmission lines of bistable units that propagate mechanical waves, specifically transition waves. The authors study the mechanics of the design, for their specific soft material, and its implementation into a variety of logic gate architectures.

Response: We appreciate the reviewer's valuable time and effort for evaluating the quality of this manuscript. We did our best to address all the comments and suggestions raised by the reviewer. Specifically, according to the reviewer's comments, we revised a sizable portion of the manuscript including *Abstract*, *Introduction*, and *Discussion*, in ways to clarify our key contribution and novelty compared to the previous studies. The following is the point-by-point responses to the reviewer's comments and suggestions, and appropriate modifications were completely reflected to the revised version of the manuscript.

#1. The basic idea of the current manuscript is to create scalable computing platforms based on propagation of mechanical bistable transition waves in soft polymers. This basic idea is already described in [Raney et al., Stable propagation of mechanical signals in soft media using stored elastic energy. *Proc. Natl. Acad. Sci. U.S.A.* 113, 9722-9727 (2016)].

The current manuscript takes this basic idea and simply creates transmission line segments to assemble logic gates together. This transmission and assembly approaches are already described in [Ion et al., Digital Mechanical Metamaterials, *Proceedings of the 2017 CHI Conference on Human Factors in Computing Systems* (New York, 2017).]

The uniqueness of the current manuscript is the mechanical characterization of their system assemblies and study of time-of-propagation for the bistable transition waves. This does not appear to bear significant scientific merit in light of the earlier publication of the basic idea and assembly approaches that are supposedly new in this manuscript. Consequently, the current manuscript represents an incremental addition to two established and published research ideas.

Response: We thank the reviewer for pointing out the two important previous papers. Firstly, we agree that Raney et al. (2016) and Ion et al. (2017) demonstrated pioneering results of snap-through instability-induced mechanical signal transmission and prototypes of mechanologics

(AND, OR gates only for Raney et al. and all functional logic gates for Ion et al.), from which the underlying design and mechanism of our integrated computing platform were inspired. However, we respectfully disagree with the reviewer on the following two points:

1) Although Raney et al. and Ion et al. describe, as the reviewer mentioned, transmission and logic gate integration, none of them demonstrates the assembly and computation through multiple networked logic gates. While Ion et al. demonstrates all types of individual mechanologies (AND, OR, NAND, and NOR gates) using “rigid signal blockers”, their system integration contains only a single AND gate with several input “signal blockers” which can hardly be used for computational autonomy in soft matter. To the best of our best knowledge, demonstrations of purely mechanical computing processes to date have been limited to a single logic gate and the corresponding fundamental logical functions. By contrast, the biggest novelty of our manuscript lies in both the **comprehensive understanding and systematic implementation of how nondispersive mechanical solitary waves can compute cascaded logical functions** by propagating through a network of structurally heterogeneous mechanical computing units, thereby achieving advanced, cascaded logical computations. It is quite well established by conventional integrated circuits that intelligence, or computation capability, is generally proportional to the number of computing units; therefore, it is quite clear that **our contribution in enhancing computational autonomy by several orders (from a single logic gate to integrated circuits) has strong scientific merit** to the field of mechanical computing.

2) The reviewer stated the uniqueness of our manuscript as “the mechanical characterization of their system assemblies and study of time-of-propagation for the bistable transition waves”. However, as we discussed earlier, the uniqueness and novelty of our manuscript is the capability of cascaded/integrated computation. The systematic characterization and optimization of all the elements allows us to achieve this functionality while maintaining a compact form factor.

In light of the above discussions, we again believe that, on top of the previous milestone works [refs. 11, 15], our study has achieved **an important advance for the design of computational metamaterial platforms that enable system-level integrated mechanical computing** for autonomous soft machines and robots. To make our main contribution more evident and differentiate the novelty of this manuscript over existing literature, we revised the manuscript as below.

Modification to the manuscript:

(We revised the *Abstract* as follows.)

Intelligent systems built upon conventional electrical architectures often face inherent hurdles for interfacing with non-electrical signals, which are common in biological organisms and soft machines. With the recent emergence of mechanical computing, completely mechanical logic computation within architected soft materials offers a new modality to formulate computational autonomy in intelligent matter without any external powering or active elements. However, advanced

processing of mechanical information in existing designs is hindered by its dissipation when interacting with networked logic gates. Here, we present a design strategy for computational soft metamaterials that provide a seamless and integrated mechanical computing interface for autonomous soft machines. Based on the systematic design and assembly of soft bistable elements, we explore how nondispersive mechanical solitary waves can compute cascaded logical functions by propagating through a network of structurally heterogeneous mechanical computing units. The developed integrated mechanical computing systems are shown to receive, transmit and compute mechanical stimuli to actuate intelligent soft machine prototypes in a seamless and integrated manner. These findings would pave the way for future intelligent robots and machines that perform computational operations between various non-electrical environmental inputs.

(We revised the *introduction* as follows.)

Intelligence embodied in biological cellular architectures often appears to mediate macroscopic dynamic responses to external mechanical stimuli. For example, mechanotransduction of sensitive plants, such as the Venus flytrap (*Aldrovanda vesiculosa*) or *Mimosa pudica* L.^{1,2}, leads to controlled opening and closing of their leaves, as the mechanical outcome of biochemical operations in response to environmental mechanical stimuli. Although seemingly binary, this behavior requires intelligent “computation” involving responsiveness to time-stamped inputs (e.g., two consecutive mechanical stimulations of the same sensory hair)³⁻⁵ or sensitivity to intensity^{6,7}, implying that computational operations between non-electrical signals engage deeply in the formulation of intelligent “living machines”.

Similarly, the design of computational responses in synthetic architected materials, termed “mechanical computing”, is of growing interest in intelligent matter and machines that perform tasks between non-electrical environmental signals, such as soft robots⁸. In this framework, computational operations are executed solely by deformation and transition dynamics of architected materials, thus featuring unconventional, electronics-free information processing with environmental signals being seamlessly coupled. To harness this advantage, ongoing efforts have been directed to supplant conventional components (e.g., gears and linkages)⁹ with soft architected materials entailing mechanical information⁸. Specifically, multistable configurations of mechanical elements that leverage snap-through instabilities of soft elastic beams¹⁰⁻¹⁷, shell¹⁸⁻²¹, or origami linkages²²⁻²⁴ abstract digitization of mechanical information. When triggered, transition behaviors between buckling modes of structurally coupled mechanical bits lead to a cascade of storage and release of elastic potential energy, which in turn results in localization^{16,25}, propagation in the form of mechanical transition waves^{11,13}, and/or logic operations^{11,15,26,27}. Efforts are now growing to employ multiple stimuli and controlling signals (e.g., electronic signals²⁸⁻³¹, magnetic

fields³²⁻³⁵, chemical stimuli^{12,36}, humidity gradients²², fluid pressure^{18,19,37,38}, etc.) that provide opportunities for further advancing the embedded processing capability, programmability, and robotic applications.

Computational autonomy as a material property per se—that is, information can be transmitted, processed, and retained solely through soft material configurations in the absence of external power sources—is the primary advantage of mechanical computing^{8,15,27}. As mechanical mechanisms encoded in material configurations account for a larger proportion in information processing, a more advanced form of completely mechanical computation (i.e., autonomous operation without any external powering or active elements) becomes critical. However, the exclusion of active elements in intelligent matter design has largely been overlooked. Indeed, demonstrations of purely mechanical computing processes have so far been limited to single logic gates and the corresponding fundamental logical functions^{12,15,22,26,27}. Although it is quite well established by conventional integrated circuits (ICs) that the computation capability is generally proportionate to the number of computing units, currently, no studies have provided meaningful insight to the design solution for computational autonomy in soft matter consisting of multiple mechanical computing units (termed “mechanologics”) intertwined with each other²². Central to this challenge is the dissipation-free transmission of computational mechanical information through a network of mechanologic clusters without the aid of external intervention or powering. A systematic understanding of mechanical signals that propagate through structurally heterogeneous mechanologics is lacking, despite both propagation in homogeneous media^{11,34,39} and operation within stand-alone mechanologics^{12,22,26,27} being relatively well studied. Accumulation of potential energy barrier (or mechanical impedance) from cascaded mechanologics acting against the propagating signals also remains an intractable problem.

Here, we present a design strategy for computational metamaterial platforms that implement system-level integrated mechanical computing for autonomous soft machines. Taking inspiration from a mature design paradigm of electronic system architectures (Fig. 1a), we demonstrate how a systematic two-dimensional (2D) assembly of engineered bistable unit elements creates a monolithic computational platform consisting of networked mechanologics, and how mechanical information, as nondispersive solitary transition waves, autonomously propagates within this architected platform to perform advanced digital computational functions (Fig. 1b). On the basis of these findings, we further generalize a design rule for computational propagation of mechanical signals via a network of computing units, and implement this governing principle in realizing system-level integrated mechanical computing. Proof-of-concept demonstrations show how the developed mechanical computing system can direct mechanical instructions across environmental inputs, computational operations and actuator modules in a monolithic form.

(Page 10, paragraph 4: We added the following paragraph.)

The primary advantage of mechanical computation platforms is that they are able to perform logical computations and retain information in the absence of external power sources⁸. This computational autonomy is attributed to completely mechanical computation according to soft material configurations (without any external powering/active elements); however, existing studies have so far been limited to demonstrations of single logic gates, limiting the number of logical functions to one^{12,15,22,26,27}. This limitation stems, as we show, from both fundamental (all information is contained in a single soliton width) and engineering (repeatably fabricating transmission lines with small soliton widths and designing compatible, rationally designed logic elements) requirements which had not been systematically investigated so far. Apart from a solitary wave-based approach, Ion et al. proposed a disconnected spring system containing rigid “signal blockers” for purely mechanical computation¹⁵. While novel, the system also suffers from the same drawback of being limited to a single logical computation. Additionally, the strategy requires, apart from the “input” signals, a separate evaluation signal which increases complexity and physical footprint while reducing the scope for autonomous actuation based on environmental signal (e.g., an AND gates requires not only both inputs to be 1, but also an additional evaluation signal to change the signal to 1; this would prevent the system to autonomously respond to two environmental inputs and instead have to wait for a manually sent evaluation signal to change the output). As a result of our methodical and parametric exploration, we are able to, for the first time, demonstrate complex yet purely mechanical integrated logic computation with autonomously propagating mechanical information. In the proposed platform, an intrinsic invariance of solitary information to the shape of excitation signals renders the computational capability robust³⁹; that is, while the snap-through response of the first element would change with different types of the input signal, the waveform of the soliton is unchanged during its propagation. Furthermore, a systematic approach for reducing the soliton width approximately to two spring elements provides a cornerstone for performing cascaded computation while maintaining a compact system form factor.

#2. The authors also suggest that their whole design philosophy is biologically inspired from plants that exhibit bistable behaviors. This is a tenuous suggestion that is not carried throughout the manuscript. There is nothing particularly biologically inspired about the logic gate assembly or methods thereof.

Response: We believe that the reviewer mistook our intention and the scope of the study. Contrary to what the reviewer mentioned, **we do not suggest a biologically inspired design philosophy** throughout the manuscript. Instead, we focused on the underlying “mechanical-to-mechanical” operations that produce intelligent responses and behaviors observed in many biological organisms: that is, “dynamic shape changes (mechanical outputs)” in response to environmental “mechanical stimuli (mechanical inputs)”. We believe that original **Figure 1** clearly implies this intention about embedded computational operations that link the mechanical input to the mechanical output.

In light of the organization of this manuscript, it is evident that **our design philosophy for integrated mechanical computing has little to do with biological inspiration from plants**. To be specific, in the *Introduction* section, we pointed out the computational operations between non-electrical signals (e.g., mechanical-to-mechanical operations) which have potential limitations in implementing them using electronic forms of computation and information processing. We then introduced the emerging field of mechanical computing in terms of intelligent matter with embedded information processing capability, and set forth the necessity of integrated mechanical computing. In the *Results* section, we explained the design strategy for optimal bistable beams, mechanical transmission lines, mechanologies (mechanical logic gates), and their cascades that ultimately enables cascaded/integrated mechanical computing. In the last demonstration part, we reinterpreted the underlying mechanical-to-mechanical operation of *M. pudica* based on the integrated mechanical computing platform, and demonstrated electronics-free autonomous soft machines that can produce intelligent responses like *M. pudica* in response to environmental mechanical stimuli. In the *Discussion* section, we summarized the key points of our study and emphasized the difference from the existing mechanologic designs. Further, we discussed the computing speed, scaling issue of our computing platform, and the capability of automated presetting and resetting process.

Given this organization of the manuscript, it turns out that the only part closely associated with biological inspiration of plants is the Mimosa-inspired autonomous soft machine as described in **Figure 6**. As well described in **Extended Data Fig. 8**, we reproduced (1) the plant cell’s osmotic gradient-driven swelling/shrinking behavior using the water (H₂O)-driven swelling/shrinking behavior of hydrogels, and (2) mechanical stimuli-induced action potential propagation for sequential shape morphing of *M. pudica* using mechanical stimuli-triggered autonomous propagation of mechanical solitary waves for sequential shape morphing of hydrogel actuators. Given the similarity of the underlying mechanisms, we do not consider this a tenuous suggestion.

Notwithstanding, according to the reviewer’s comment, we did our best to improve the clarity by deleting the figures and descriptions that might cause misunderstanding. Please see the below modifications.

Modification to the manuscript:

(Figure 1 was changed.)

(We revised the Abstract as below.)

~~Many intelligent behaviors of biological organisms are the outcome of seamless, integrated operations between various types of biological signals and environmental stimuli.~~ In contrast, artificial intelligent systems built upon conventional electrical architectures often face inherent hurdles for interfacing with non-electrical signals, which are common in soft machines. Here, we present a design strategy for computational soft metamaterials that provide a seamless and integrated mechanical computing interface for autonomous soft machines. Based on the systematic design and assembly of soft bistable elements, we explore how mechanical solitary waves can create computational mechano-intelligence by propagating through a network of structurally heterogeneous mechanical computing units. The developed integrated mechanical computing systems are shown to receive, transmit and compute mechanical stimuli to actuate intelligent soft machine prototypes in a seamless and integrated manner. These findings would pave the way for future intelligent robots and machines that perform computational operations between various non-electrical environmental inputs.

⇒ **Intelligent systems built upon conventional electrical architectures often face inherent hurdles for interfacing with non-electrical signals, which are common in biological organisms and soft machines. With the recent emergence of mechanical computing, completely mechanical logic computation within architected soft materials offers a new modality to formulate computational autonomy in intelligent matter without any external powering or active elements. However, advanced processing of mechanical information in existing designs is hindered by its dissipation when interacting with networked logic gates.** Here, we present a design strategy for computational soft metamaterials

that provide a seamless and integrated mechanical computing interface for autonomous soft machines. Based on the systematic design and assembly of soft bistable elements, we explore how **nondispersive** mechanical solitary waves **can compute cascaded logical functions** by propagating through a network of structurally heterogeneous mechanical computing units. The developed integrated mechanical computing systems are shown to receive, transmit and compute mechanical stimuli to actuate intelligent soft machine prototypes in a seamless and integrated manner. These findings would pave the way for future intelligent robots and machines that perform computational operations between various non-electrical environmental inputs.

(Page 1, paragraph 1: We revised the paragraph as follows.)

Intelligence embodied in biological lifeforms often appears to mediate their dynamic responses to external mechanical stimuli. For example, mechanotransduction of sensitive plants, such as the Venus flytrap (*Aldrovanda vesiculosa*) or *Mimosa pudica* L.^{1,2}, leads to controlled opening and closing of their leaves, as the mechanical outcome of biochemical operations in response to environmental mechanical stimuli. Although seemingly binary, this behavior requires intelligent “computation” involving responsiveness to time-stamped inputs (e.g., two consecutive mechanical stimulations of the same sensory hair)³⁻⁵ or sensitivity to intensity^{6,7}. The internal operating circuit that drives such intelligent responses leads to autonomous, integrated living machines in which sensing, computation, feedback and actuation are seamlessly incorporated (Fig. 1a-i)).

⇒ Intelligence embodied in **biological cellular architectures** often appears to mediate **macroscopic** dynamic responses to external mechanical stimuli. For example, mechanotransduction of sensitive plants, such as the Venus flytrap (*Aldrovanda vesiculosa*) or *Mimosa pudica* L.^{1,2}, leads to controlled opening and closing of their leaves, as the mechanical outcome of biochemical operations in response to environmental mechanical stimuli. Although seemingly binary, this behavior requires intelligent “computation” involving responsiveness to time-stamped inputs (e.g., two consecutive mechanical stimulations of the same sensory hair)³⁻⁵ or sensitivity to intensity^{6,7}, **implying that computational operations between non-electrical signals engage deeply in the formulation of intelligent “living machines”**.

#3. Overall this manuscript does not appear to have great scientific novelty in light of prior published work.

Response: Architected, intelligent matter with embedded information processing capability has attracted increasing interest for autonomous deformation analysis, control, sensing and actuation, without traditional (e.g., electrical or pneumatic) control modalities. In this sense, we expect that the goal of mechanical computing is to advance the intelligence of artificial matter based on high-level “mechano-intelligence”. In light of conventional electronic integrated circuit design, it is certain that the high-level mechano-intelligence can be realized by (i) cascades (or networks) of mechanical computing devices (i.e., mechanologics) and (ii) the systematic investigation of the behavior of nondispersive mechanical information passing through them. This is exactly what we studied and presented in this manuscript. We are also fully aware that prior published works, including Raney et al. [ref. 11] and Ion et al. [ref. 15], have laid the groundwork for mechanical computing in terms of mechanologic design and transmissive signal propagation. However, it is an undeniable fact that **they did not overcome the above challenges for high-level mechano-intelligence which are not only scientifically significant, but also vital to progress in the relevant field.** In this study, we developed a design strategy of mechanologics that enable compact, seamless and cascaded mechanical computing, and finally established a systematic approach to understand the behavior of computational mechanical transition waves propagating through a network of the mechanologics. The result was highlighted by the first development of autonomous soft machines where the internal operating circuit for high-level information processing is implemented solely by snap-through-based autonomous deformation and transition of architected soft materials. Consequently, we strongly believe that **our achievement is an important step towards computational metamaterials with high-level mechano-intelligence, representing great scientific and engineering novelties that are clearly distinctive from prior published studies.** According to the reviewer’s comment, we revised a sizable portion of the manuscript to clarify and emphasize our novelty. Please see the answer of the comment #1 for detailed modifications.

Reviewer #2

The authors propose a systematic design strategy for computational metamaterial platforms capable of system-level integrated mechanical computing in the context of autonomous soft machines. They show how a systematic 2D assembly of engineered bistable unit elements can create a monolithic computational platform consisting of networked mechanical computing devices and how mechanical information, in the form of nondispersive solitary transition waves, propagates within this platform to perform digital computational functions such as NOT, AND, OR.

The idea is not new but the specific implementations seem to have some degree of originality. Indeed, the use of bistable elements in programmable metamaterials is not new but its systematic use to create the proposed architectures seems to be new. The authors have put significant efforts into showing various architectures and functions. The paper is well-written and deserves consideration in *Nature Communications*. However, a major revision is necessary.

Response: We thank the reviewer recommending consideration of this manuscript and for appreciating the systematic analysis, novelty, and demonstrations of various new architectures and function that we show. We also agree with the reviewer about explicitly highlighting the novelty of this work and have modified the *Introduction* and *Discussion* sections to make it easier to perceive. The following is the point-by-point responses to the reviewer's comments and suggestions, and appropriate modifications were completely reflected to the revised version of the manuscript.

#1. The authors have to state more precisely what is new in their implementation with respect to the wide literature on mechanical computing.

Response: We thank the review for this comment, and have taken this opportunity to highlight the novelty of the manuscript and emphasize the advances from the current state-of-the-art we demonstrate in the field of mechanical computing. Researchers agree that computational autonomy as a material property per se—that is, information can be transmitted, processed, and retained solely through soft material configurations in the absence of external power sources—is the primary advantage of mechanical computing [Ref. 8]. There is no doubt that as mechanical mechanisms encoded in material configurations account for a larger proportion in information processing, a more advanced form of completely mechanical computation (i.e., autonomous operation without any external powering or active elements) becomes critical. In the revised manuscript, according to the reviewer's comment, we have more precisely pointed out what the major unresolved challenge is for this advanced, purely mechanical computing—which we believe is that completely mechanical logic computation in existing designs is hindered by the dissipation of mechanical information when interacting with networked logic gates. We have further made clear what is lacking and what the central challenges are to address this issue.

Taking the above discussion into account, we have modified both the *introduction* and the *discussion* sections to precisely point out what is new in our implementations.

Modification to the manuscript:

(We revised the *Abstract* as follows.)

Intelligent systems built upon conventional electrical architectures often face inherent hurdles for interfacing with non-electrical signals, which are common in biological organisms and soft machines. With the recent emergence of mechanical computing, completely mechanical logic computation within architected soft materials offers a new modality to formulate computational autonomy in intelligent matter without any external powering or active elements. However, advanced processing of mechanical information in existing designs is hindered by its dissipation when interacting with networked logic gates. Here, we present a design strategy for computational soft metamaterials that provide a seamless and integrated mechanical computing interface for autonomous soft machines. Based on the systematic design and assembly of soft bistable elements, we explore how nondispersive mechanical solitary waves can compute cascaded logical functions by propagating through a network of structurally heterogeneous mechanical computing units. The developed integrated mechanical computing systems are shown to receive, transmit and compute mechanical stimuli to actuate intelligent soft machine prototypes in a seamless and integrated manner. These findings would pave the way for future intelligent robots and machines that perform computational operations between various non-electrical environmental inputs.

(We revised the *introduction* as follows.)

Intelligence embodied in biological cellular architectures often appears to mediate macroscopic dynamic responses to external mechanical stimuli. For example, mechanotransduction of sensitive plants, such as the Venus flytrap (*Aldrovanda vesiculosa*) or *Mimosa pudica* L.^{1,2}, leads to controlled opening and closing of their leaves, as the mechanical outcome of biochemical operations in response to environmental mechanical stimuli. Although seemingly binary, this behavior requires intelligent “computation” involving responsiveness to time-stamped inputs (e.g., two consecutive mechanical stimulations of the same sensory hair)³⁻⁵ or sensitivity to intensity^{6,7}, implying that computational operations between non-electrical signals engage deeply in the formulation of intelligent “living machines”.

Similarly, the design of computational responses in synthetic architected materials, termed “mechanical computing”, is of growing interest in intelligent matter and machines that perform tasks between non-electrical environmental signals, such as soft robots⁸. In this framework, computational operations are executed solely by deformation and transition dynamics of architected materials, thus featuring unconventional, electronics-free information processing with environmental signals being seamlessly coupled. To harness this advantage, ongoing efforts have been directed to supplant conventional components (e.g.,

gears and linkages)⁹ with soft architected materials entailing mechanical information⁸. Specifically, multistable configurations of mechanical elements that leverage snap-through instabilities of soft elastic beams¹⁰⁻¹⁷, shell¹⁸⁻²¹, or origami linkages²²⁻²⁴ abstract digitization of mechanical information. When triggered, transition behaviors between buckling modes of structurally coupled mechanical bits lead to a cascade of storage and release of elastic potential energy, which in turn results in localization^{16,25}, propagation in the form of mechanical transition waves^{11,13}, and/or logic operations^{11,15,26,27}. Efforts are now growing to employ multiple stimuli and controlling signals (e.g., electronic signals²⁸⁻³¹, magnetic fields³²⁻³⁵, chemical stimuli^{12,36}, humidity gradients²², fluid pressure^{18,19,37,38}, etc.) that provide opportunities for further advancing the embedded processing capability, programmability, and robotic applications.

Computational autonomy as a material property per se—that is, information can be transmitted, processed, and retained solely through soft material configurations in the absence of external power sources—is the primary advantage of mechanical computing^{8,15,27}. As mechanical mechanisms encoded in material configurations account for a larger proportion in information processing, a more advanced form of completely mechanical computation (i.e., autonomous operation without any external powering or active elements) becomes critical. However, the exclusion of active elements in intelligent matter design has largely been overlooked. Indeed, demonstrations of purely mechanical computing processes have so far been limited to single logic gates and the corresponding fundamental logical functions^{12,15,22,26,27}. Although it is quite well established by conventional integrated circuits (ICs) that the computation capability is generally proportionate to the number of computing units, currently, no studies have provided meaningful insight to the design solution for computational autonomy in soft matter consisting of multiple mechanical computing units (termed “mechanologics”) intertwined with each other²². Central to this challenge is the dissipation-free transmission of computational mechanical information through a network of mechanologic clusters without the aid of external intervention or powering. A systematic understanding of mechanical signals that propagate through structurally heterogeneous mechanologics is lacking, despite both propagation in homogeneous media^{11,34,39} and operation within stand-alone mechanologics^{12,22,26,27} being relatively well studied. Accumulation of potential energy barrier (or mechanical impedance) from cascaded mechanologics acting against the propagating signals also remains an intractable problem.

Here, we present a design strategy for computational metamaterial platforms that implement system-level integrated mechanical computing for autonomous soft machines. Taking inspiration from a mature design paradigm of electronic system architectures (Fig. 1a), we demonstrate how a systematic two-dimensional (2D) assembly of engineered bistable unit elements creates a monolithic computational platform consisting of networked mechanologics, and

how mechanical information, as nondispersive solitary transition waves, autonomously propagates within this architected platform to perform advanced digital computational functions (Fig. 1b). On the basis of these findings, we further generalize a design rule for computational propagation of mechanical signals via a network of computing units, and implement this governing principle in realizing system-level integrated mechanical computing. Proof-of-concept demonstrations show how the developed mechanical computing system can direct mechanical instructions across environmental inputs, computational operations and actuator modules in a monolithic form.

(Page 10, paragraph 4: We added the following paragraph.)

The primary advantage of mechanical computation platforms is that they are able to perform logical computations and retain information in the absence of external power sources⁸. This computational autonomy is attributed to completely mechanical computation according to soft material configurations (without any external powering/active elements); however, existing studies have so far been limited to demonstrations of single logic gates, limiting the number of logical functions to one^{12,15,22,26,27}. This limitation stems, as we show, from both fundamental (all information is contained in a single soliton width) and engineering (repeatably fabricating transmission lines with small soliton widths and designing compatible, rationally designed logic elements) requirements which had not been systematically investigated so far. Apart from a solitary wave-based approach, Ion et al. proposed a disconnected spring system containing rigid “signal blockers” for purely mechanical computation¹⁵. While novel, the system also suffers from the same drawback of being limited to a single logical computation. Additionally, the strategy requires, apart from the “input” signals, a separate evaluation signal which increases complexity and physical footprint while reducing the scope for autonomous actuation based on environmental signal (e.g., an AND gates requires not only both inputs to be 1, but also an additional evaluation signal to change the signal to 1; this would prevent the system to autonomously respond to two environmental inputs and instead have to wait for a manually sent evaluation signal to change the output). As a result of our methodical and parametric exploration, we are able to, for the first time, demonstrate complex yet purely mechanical integrated logic computation with autonomously propagating mechanical information. In the proposed platform, an intrinsic invariance of solitary information to the shape of excitation signals renders the computational capability robust³⁹; that is, while the snap-through response of the first element would change with different types of the input signal, the waveform of the soliton is unchanged during its propagation. Furthermore, a systematic approach for reducing the soliton width approximately to two spring elements provides a cornerstone for performing cascaded computation while maintaining a compact system form factor.

#2. The overall idea of mechanical computing is interesting but the speed barrier associated with mechanical transduction between adjacent bistable elements is not discussed.

Response: We thank the reviewer for bringing up this important point. The major determinant of the computing speed in the proposed platform is basically twofold: (i) the speed of the “computing process” itself, and (ii) the transmission speed of mechanical solitary waves through the transmission line. Focusing on the latter to address the reviewer’s point, we identify that the transmission speed is determined by the combination of energy-release behavior of each bistable beam (thus, the bistability profile) and conventional elastic wave velocity ($v = \sqrt{E/\rho} \approx \sqrt{k/\rho}$, where E is the Young’s modulus, ρ is the density, and k is the spring constant or stiffness; the approximation is valid under the linear elastic regime). Assuming the snap-through transformation of bistable elements being a spontaneous motion, the timescale of entire computation processes is governed by elastic wave velocity $\sim\sqrt{k}$. This investigation suggests that the speed barrier associated with mechanical transduction between adjacent bistable elements varies by orders of magnitude with respect to k ($5 - 200 \text{ mm s}^{-1}$ for our experiments with data not included and $1 - 20 \text{ m s}^{-1}$ for the previous study [Ref. 11]). Therefore, increasing the spring stiffness (k) would be an option to improve the computing speed, but a close positive correlation between k and $N_{soliton}$ (Fig. 2e) may pose a trade-off between wave speed and system compactness (i.e., integration density). As a result, we chose an optimal value of k of 180 N m^{-1} which leads to $N_{soliton}$ of ~ 2 and transmission speed of $8-10 \text{ mm s}^{-1}$ (Extended Data Fig. 4b). To include this discussion, we have modified the manuscript as below.

Modification to the manuscript:

(Page 11, paragraph 3: We revised the paragraph as follows.)

The computing speed and scaling-down issues of the proposed mechanical computing system are another major challenge for practical use. First, the factors governing the computing speed are basically twofold: (i) the speed of the “computing process” itself, and (ii) the transmission speed of mechanical solitary waves through the transmission line. The former factor, which is interpreted as sequential transition behaviors between buckling modes of the bistable elements that comprise mechanologics, is heavily reliant on the bistable properties and the corresponding input/output characteristics of mechanologics; the experimentally obtained values (normalized by the lattice constant, a) were $0.06 - 0.33 \text{ s}^{-1}$ for the cases of NOT gates (see Fig. 3e). On the other hand, the transmission speed determined by the combination of energy-release behavior of each bistable beam (thus, the bistability profile) and conventional elastic wave velocity ($v = \sqrt{E/\rho} \approx \sqrt{k/\rho}$, where E is the Young’s modulus and ρ is the density; the approximation is valid under the linear elastic regime). Assuming the snap-through transformation of bistable elements being a spontaneous motion, the timescale of entire computation processes ($\sim v^{-1} \sim k^{-1/2}$) is governed by elastic wave velocity. This

investigation suggests that the speed barrier associated with mechanical transduction between adjacent bistable elements varies by orders of magnitude with respect to k (5–200 mm s⁻¹ for our experiments, data not included)¹¹. Therefore, increasing the spring stiffness (k) would be an option to improve the computing speed, but a close positive correlation between k and $N_{soliton}$ (Fig. 2e) may pose a trade-off between wave speed and system compactness (i.e., integration density). We chose an optimal value of k of 180 N m⁻¹ which leads to $N_{soliton}$ of ~ 2 and transmission speed of 8–10 mm s⁻¹ (Extended Data Fig. 4b), fulfilling an acceptable computation speed while maintaining a compact form factor. Similarly, given the normalized snap-through force and potential energy of a unit bistable element being scale-independent, $\sim (t/a)^3$ (see Methods), a scaling-down issue is governed by elastic linkage structures whose spring constant (k) is linearly proportional to the length scale. In this regard, selective material assignment (e.g., using stiffer materials for the linkage parts) could be considered as a potential solution for both of these issues of computing dynamics, speed and scaling.

#3. An important issue is the role played by impulsive or general dynamics excitations. If architected metamaterials made of bistable elements such as those here proposed are subject to dynamic excitations introduced at different points along the chains, the mechanical response can become rather chaotic. What happens in terms of computational capability?

Response: We thank the reviewer for bringing up this important point. One of the advantages of using a series of bistable elements through which a transition wave can propagate is that the waveform of the soliton, which contains all the information, is input invariant; i.e., it does not depend on the shape of the input signal. This input-invariance, along with the fact that information transmission through transition waves are lossless (i.e., the output does not depend on the distance of propagation) is a major reason why we believe transition waves are especially suited for mechanical computing. In spite of these advantages, as the reviewer correctly pointed out, adding dynamic excitations to the middle of the chain may cause some unwanted results; but this is true as well for all types of signal propagation systems. Even in electromagnetic or electrical systems, adding external noise during signal transmission can corrupt the signal; and similar to electromagnetic shielding, mechanical damping could be used to help reduce the effect of external noise. We have included these points to make our discussion comprehensive.

Modification to the manuscript:

(Page 10, paragraph 4: We added the following paragraph.)

The primary advantage of mechanical computation platforms is that they are able to perform logical computations and retain information in the absence of external power sources⁸. This computational autonomy is attributed to completely

mechanical computation according to soft material configurations (without any external powering/active elements); however, existing studies have so far been limited to demonstrations of single logic gates, limiting the number of logical functions to one^{12,15,22,26,27}. This limitation stems, as we show, from both fundamental (all information is contained in a single soliton width) and engineering (repeatably fabricating transmission lines with small soliton widths and designing compatible, rationally designed logic elements) requirements which had not been systematically investigated so far. Apart from a solitary wave-based approach, Ion et al. proposed a disconnected spring system containing rigid “signal blockers” for purely mechanical computation¹⁵. While novel, the system also suffers from the same drawback of being limited to a single logical computation. Additionally, the strategy requires, apart from the “input” signals, a separate evaluation signal which increases complexity and physical footprint while reducing the scope for autonomous actuation based on environmental signal (e.g., an AND gates requires not only both inputs to be 1, but also an additional evaluation signal to change the signal to 1; this would prevent the system to autonomously respond to two environmental inputs and instead have to wait for a manually sent evaluation signal to change the output). As a result of our methodical and parametric exploration, we are able to, for the first time, demonstrate complex yet purely mechanical integrated logic computation with autonomously propagating mechanical information. **In the proposed platform, an intrinsic invariance of solitary information to the shape of excitation signals renders the computational capability robust³⁹; that is, while the snap-through response of the first element would change with different types of the input signal, the waveform of the soliton is unchanged during its propagation.** Furthermore, a systematic approach for reducing the soliton width approximately to two spring elements provides a cornerstone for performing cascaded computation while maintaining a compact system form factor.

#4. Please improve the general quality of the figures. Sometimes it is really hard to read the labels or the numbers given the high density of subfigures.

Response: We thank the reviewer for this kind comment. According to the reviewer’s suggestion, we have improved the quality and clearness of all figures. Please refer to the revised manuscript.

Reviewer #3

The paper proposes a technique for implementing mechanical computers that take inspiration from biological systems. The paper describes an novel computing cell architecture, analytical model for the activation thresholds and mechanical and numerical tests have been conducted. Finally, it demonstrates the viability of the approach by replicating the computational characteristics of a Mimosa plant.

The paper is well written and exhibits a high rigor that deepens the understanding the proposed structures and provides a solid foundation to build on for future work.

Overall, I would recommend moving forward with this manuscript, as it is a valuable step toward implementing mechanical, biologically inspired computing systems.

Response: We appreciate the reviewer's valuable time and effort for evaluating the quality of this manuscript. We thank the reviewer for his words of support and for appreciating the rigor of our manuscript. We agree with the reviewer about explicitly highlighting the novelty of this work and have modified the *Introduction* and *Discussion* sections to make it easier to perceive. The following is the point-by-point responses to the reviewer's comments and suggestions, and appropriate modifications were completely reflected to the revised version of the manuscript.

#1. The paper gives a broad overview over approaches in other disciplines for implementing mechanical computation. I am missing a short discussion of the delta of this work over e.g. "Digital Mechanical Metamaterials" Ion et al.. Especially the advantages of propagating signals using the solitary wave approach over disconnected springs should be highlighted. The authors describe a "lack of systematic investigations on cascaded and integrated mechanical computing" in the related work. While there is always room for improvement, it seems to me like a overly generalizing statement. I would prefer, if the authors state which systematic investigations they are contributing instead.

Response: We thank the reviewer for bringing up this important point. We agree with the reviewer that the original version of the introduction section, especially for the presentation of central challenges, was written like an overly generalizing statement, and that a clear discussion of the delta of our achievement over the previous important ones (e.g., Raney et al. [ref. 25] and Ion et al. [ref. 15]) was lacking. Researchers agree that computational autonomy as a material property per se—that is, information can be transmitted, processed, and retained solely through soft material configurations in the absence of external power sources—is the primary advantage of mechanical computing [Ref. 8]. There is no doubt that as mechanical mechanisms encoded in material configurations account for a larger proportion in information processing, a more advanced form of completely mechanical computation (i.e., autonomous operation without any external powering or active elements) becomes critical. However, advanced processing of mechanical information in existing designs is hindered by its dissipation when interacting with networked logic gates. Our key contribution to the mechanical computing society lies in both the **comprehensive understanding and systematic implementation of how nondispersive mechanical solitary waves can compute cascaded logical functions by**

propagating through a network of structurally heterogeneous mechanical computing units.

Taking the above discussion into account, according to the reviewer's suggestion, we revised the *Abstract*, *Introduction*, and *Discussion* in ways that give shape to what the major unresolved challenge is for advanced, purely mechanical computing, what is lacking and what the central hurdles are to address this issue.

Modification to the manuscript:

(We revised the *Abstract* as follows.)

Intelligent systems built upon conventional electrical architectures often face inherent hurdles for interfacing with non-electrical signals, which are common in biological organisms and soft machines. With the recent emergence of mechanical computing, completely mechanical logic computation within architected soft materials offers a new modality to formulate computational autonomy in intelligent matter without any external powering or active elements. However, advanced processing of mechanical information in existing designs is hindered by its dissipation when interacting with networked logic gates. Here, we present a design strategy for computational soft metamaterials that provide a seamless and integrated mechanical computing interface for autonomous soft machines. Based on the systematic design and assembly of soft bistable elements, we explore how nondispersive mechanical solitary waves can compute cascaded logical functions by propagating through a network of structurally heterogeneous mechanical computing units. The developed integrated mechanical computing systems are shown to receive, transmit and compute mechanical stimuli to actuate intelligent soft machine prototypes in a seamless and integrated manner. These findings would pave the way for future intelligent robots and machines that perform computational operations between various non-electrical environmental inputs.

(We revised the *introduction* as follows.)

Intelligence embodied in biological cellular architectures often appears to mediate macroscopic dynamic responses to external mechanical stimuli. For example, mechanotransduction of sensitive plants, such as the Venus flytrap (*Aldrovanda vesiculosa*) or *Mimosa pudica* L.^{1,2}, leads to controlled opening and closing of their leaves, as the mechanical outcome of biochemical operations in response to environmental mechanical stimuli. Although seemingly binary, this behavior requires intelligent “computation” involving responsiveness to time-stamped inputs (e.g., two consecutive mechanical stimulations of the same sensory hair)³⁻⁵ or sensitivity to intensity^{6,7}, implying that computational operations between non-electrical signals engage deeply in the formulation of intelligent “living machines”.

Similarly, the design of computational responses in synthetic architected materials, termed “mechanical computing”, is of growing interest in intelligent matter and machines that perform tasks between non-electrical environmental signals, such as soft robots⁸. In this framework, computational operations are executed solely by deformation and transition dynamics of architected materials, thus featuring unconventional, electronics-free information processing with environmental signals being seamlessly coupled. To harness this advantage, ongoing efforts have been directed to supplant conventional components (e.g., gears and linkages)⁹ with soft architected materials entailing mechanical information⁸. Specifically, multistable configurations of mechanical elements that leverage snap-through instabilities of soft elastic beams¹⁰⁻¹⁷, shell¹⁸⁻²¹, or origami linkages²²⁻²⁴ abstract digitization of mechanical information. When triggered, transition behaviors between buckling modes of structurally coupled mechanical bits lead to a cascade of storage and release of elastic potential energy, which in turn results in localization^{16,25}, propagation in the form of mechanical transition waves^{11,13}, and/or logic operations^{11,15,26,27}. Efforts are now growing to employ multiple stimuli and controlling signals (e.g., electronic signals²⁸⁻³¹, magnetic fields³²⁻³⁵, chemical stimuli^{12,36}, humidity gradients²², fluid pressure^{18,19,37,38}, etc.) that provide opportunities for further advancing the embedded processing capability, programmability, and robotic applications.

Computational autonomy as a material property per se—that is, information can be transmitted, processed, and retained solely through soft material configurations in the absence of external power sources—is the primary advantage of mechanical computing^{8,15,27}. As mechanical mechanisms encoded in material configurations account for a larger proportion in information processing, a more advanced form of completely mechanical computation (i.e., autonomous operation without any external powering or active elements) becomes critical. However, the exclusion of active elements in intelligent matter design has largely been overlooked. Indeed, demonstrations of purely mechanical computing processes have so far been limited to single logic gates and the corresponding fundamental logical functions^{12,15,22,26,27}. Although it is quite well established by conventional integrated circuits (ICs) that the computation capability is generally proportionate to the number of computing units, currently, no studies have provided meaningful insight to the design solution for computational autonomy in soft matter consisting of multiple mechanical computing units (termed “mechanologics”) intertwined with each other²². Central to this challenge is the dissipation-free transmission of computational mechanical information through a network of mechanologic clusters without the aid of external intervention or powering. A systematic understanding of mechanical signals that propagate through structurally heterogeneous mechanologics is lacking, despite both propagation in homogeneous media^{11,34,39} and operation within stand-alone mechanologics^{12,22,26,27} being relatively well studied. Accumulation of potential energy barrier (or mechanical impedance) from

cascaded mechanologics acting against the propagating signals also remains an intractable problem.

Here, we present a design strategy for computational metamaterial platforms that implement system-level integrated mechanical computing for autonomous soft machines. Taking inspiration from a mature design paradigm of electronic system architectures (Fig. 1a), we demonstrate how a systematic two-dimensional (2D) assembly of engineered bistable unit elements creates a monolithic computational platform consisting of networked mechanologics, and how mechanical information, as nondispersive solitary transition waves, autonomously propagates within this architected platform to perform advanced digital computational functions (Fig. 1b). On the basis of these findings, we further generalize a design rule for computational propagation of mechanical signals via a network of computing units, and implement this governing principle in realizing system-level integrated mechanical computing. Proof-of-concept demonstrations show how the developed mechanical computing system can direct mechanical instructions across environmental inputs, computational operations and actuator modules in a monolithic form.

(Page 10, paragraph 4: We added the following paragraph.)

The primary advantage of mechanical computation platforms is that they are able to perform logical computations and retain information in the absence of external power sources⁸. This computational autonomy is attributed to completely mechanical computation according to soft material configurations (without any external powering/active elements); however, existing studies have so far been limited to demonstrations of single logic gates, limiting the number of logical functions to one^{12,15,22,26,27}. This limitation stems, as we show, from both fundamental (all information is contained in a single soliton width) and engineering (repeatably fabricating transmission lines with small soliton widths and designing compatible, rationally designed logic elements) requirements which had not been systematically investigated so far. Apart from a solitary wave-based approach, Ion et al. proposed a disconnected spring system containing rigid “signal blockers” for purely mechanical computation¹⁵. While novel, the system also suffers from the same drawback of being limited to a single logical computation. Additionally, the strategy requires, apart from the “input” signals, a separate evaluation signal which increases complexity and physical footprint while reducing the scope for autonomous actuation based on environmental signal (e.g., an AND gates requires not only both inputs to be 1, but also an additional evaluation signal to change the signal to 1; this would prevent the system to autonomously respond to two environmental inputs and instead have to wait for a manually sent evaluation signal to change the output). As a result of our methodical and parametric exploration, we are able to, for the first time, demonstrate complex yet purely mechanical integrated

logic computation with autonomously propagating mechanical information. In the proposed platform, an intrinsic invariance of solitary information to the shape of excitation signals renders the computational capability robust³⁹; that is, while the snap-through response of the first element would change with different types of the input signal, the waveform of the soliton is unchanged during its propagation. Furthermore, a systematic approach for reducing the soliton width approximately to two spring elements provides a cornerstone for performing cascaded computation while maintaining a compact system form factor.

REVIEWER COMMENTS

Reviewer #2 (Remarks to the Author):

The authors have addressed the points in a very accurate and professional way. I believe that the paper can be recommended for publication as is.

Reviewer #3 (Remarks to the Author):

I welcome the changes the authors made over the last version of the manuscript.

I find the clarifications regarding the contribution over the related work convincing: This contributes a more types of logic gates as well as a thorough understanding of the activation energy and other criteria necessary for building on this work.

Hence, I would argue for moving forward with this manuscript.

Reviewer #4 (Remarks to the Author):

This manuscript describes the design and operation of nondispersive solitary waves integrated with mechanical logic units. The authors demonstrate signal propagation through transmission lines and multiple logic gates. They also provide design rules for how create logic systems based on their concepts.

As the first round reviewers point out, other papers established the basic building blocks of this work: Raney et al. demonstrated transmission lines, and Ion et al. show logic integration. Furthermore, Librandi et al (not cited in the paper) [Librandi, G., Tubaldi, E. & Bertoldi, K. Programming nonreciprocity and reversibility in multistable mechanical metamaterials. *Nat Commun* **12**, 3454 (2021).] already established the fundamental concept of programmable non-reciprocity by tuning the differential energy wells for bistable elements – which is presented in

Fig. 3. The novelty here is that the authors show the ability to propagate through multiple levels of logic gates. This is an incremental, but important advance.

In their introduction, the authors claim that “advanced processing of mechanical information in existing designs is hindered by its dissipation when interaction with networked logic gates.” This is an interesting claim, however a methodical exploration within the paper is not clear. Instead, the authors identify their system as “seamless” and “integrated” which are important attributes of an actualized system, but do not strongly indicate novel scientific work.

The detailed description of the biological example from which the authors drew inspiration distracted from the central claim of the paper: demonstrating multi-layered signal propagation. The authors could pare back the biological discussion and instead add detail to other sections.

Response Letter

We would like to thank the reviewers for their valuable, detailed and constructive feedback, which has helped to strengthen our manuscript. We have addressed all comments and suggestions provided by the reviewers point by point below. The changes made to the manuscript are highlighted in red in the revised manuscript.

Reviewer #2

The authors have addressed the points in a very accurate and professional way. I believe that the paper can be recommended for publication as is.

Response: We greatly appreciate the reviewer's recommendation for publication.

Reviewer #3

I welcome the changes the authors made over the last version of the manuscript.

I find the clarifications regarding the contribution over the related work convincing: This contributes a more types of logic gates as well as a thorough understanding of the activation energy and other criteria necessary for building on this work.

Hence, I would argue for moving forward with this manuscript.

Response: We thank the reviewer for appreciating the novelty of our study, and also for recommending our manuscript for publication.

Reviewer #4

This manuscript describes the design and operation of nondispersive solitary waves integrated with mechanical logic units. The authors demonstrate signal propagation through transmission lines and multiple logic gates. They also provide design rules for how create logic systems based on their concepts.

Response: We appreciate the reviewer’s valuable time and effort for evaluating the quality of this manuscript. We specially thank the reviewer for appreciating our key contribution on transition wave-based cascaded mechanical computing through multi-level logic gates as an important advance. We did our best to address all the comments and suggestions raised by the reviewer, and accordingly revised a sizable portion of the manuscript including *Abstract*, *Introduction*, *Results* and *Figures*, in ways to clarify our key contribution/novelty and methodical exploration that enables it, in comparison with the previous studies. The following is the point-by-point responses to the reviewer’s comments and suggestions, and appropriate modifications were completely reflected to the revised version of the manuscript. Note that all Extended Data Figures (from 1 to 10) were moved to Supplementary Figures and then re-numbered in order to meet the standard format of *Nature Communications*.

#1. As the first round reviewers point out, other papers established the basic building blocks of this work: Raney et al. demonstrated transmission lines, and Ion et al. show logic integration. Furthermore, Librandi et al (not cited in the paper) [Librandi, G., Tubaldi, E. & Bertoldi, K. “Programming nonreciprocity and reversibility in multistable mechanical metamaterials.” *Nat, Commun*, 12, 3454 (2021).] already established the fundamental concept of programmable non-reciprocity by tuning the differential energy wells for bistable elements – which is presented in Fig. 3. The novelty here is that the authors show the ability to propagate through multiple levels of logic gates. This is an incremental, but important advance.

Response: We thank the reviewer for this comment. As we clarified in the 1st-round revision to other reviewers, we are fully aware that our study is built upon a few important papers, among which include Raney et al. [ref. 11], Ion et al. [ref. 15], and Librandi et al [ref. R4.1]. In particular, as the reviewer pointed out, programmable non-reciprocity and reversibility in soft mechanical computing—which can be achieved by metamaterial design with tunable potential wells, as suggested by Librandi [ref. R4.1], and/or stimuli-responsive materials [refs. 29-39]—have laid a foundation for repetitive computation and/or autonomous resetting. We thank the reviewer for informing us of this important paper that was previously omitted from the reference list.

While Librandi et al. did demonstrate tuning the differential energy wells for a 1D array of bistable arches, we would like to make clear that the originality of our study lies in how to design the ability of mechanical transition waves to propagate through multiple levels of logic gates, namely ‘mechanologics’, with less emphasis on the non-reciprocity or reversibility (in the third paragraph of the *Discussion* section, we have already discussed the current limitation of the one-way propagation). Instead, our achievements advanced the

capability to quantify and tune decisive energy components of both mechanical transmission lines and mechanologics—for example, the input energy barrier (U_b) and the transmissive energy delivered to the output end of the transmission line (U_{out})—both of which engage deeply in solitary wave-based mechanical computation. Based on these investigations, we succeeded in generalizing a design solution to the autonomous propagation through networked mechanologic clusters. We believe that the design, characterization and subsequent optimization of the energies of cascaded computing systems, following our developed design guideline, are the elemental in developing and fabricating integrated, multi-level computational circuits for mechanical signals. Taking the above discussion into account, we have modified the manuscript as below, with inclusion of the paper by Librandi et al. into the reference list:

Reference

R4.1. Librandi, G., Tubaldi, E. & Bertoldi, K. Programming nonreciprocity and reversibility in multistable mechanical metamaterials. *Nature Commun.* **12**, 3454 (2021).

Modification to the manuscript:

(In page 3, 1st paragraph: we added the reference #26.)

(...) When triggered, transition behaviors between buckling modes of structurally coupled mechanical bits lead to a cascade of storage and release of elastic potential energy, which in turn results in localization^{16,25}, propagation in the form of **mechanical transition waves**^{11,13,26}, and/or logic operations^{11,15,27,28}. (...)

(In page 11, 2nd paragraph: we revised the paragraph as follows.)

While this study focuses on the fundamental design strategy for enhancing computational autonomy in completely mechanical soft-matter systems, real-world applications should entail the capability of automated presets of bistable elements and devices. Although the current **non-reciprocal** bistable design only permitted spontaneous one-way propagation (Extended Data Fig. 1)¹¹, **the elaborate design of energy landscapes in bistable elements**²⁶ and/or the inclusion of active control modalities, such as pneumatic flows^{18,19,38,39}, electronics²⁹⁻³², (electro-)magnets³³⁻³⁶ or other environmental inputs^{12,37}, could provide room for reversal of energetically favorable behaviors. Future work in this direction will include the development of active, heterogeneous building blocks that contain stimuli-responsive (e.g., magnetic fields, water, pH, etc.), shape-changing materials (see Supplementary Fig. 9).

(We added the **ref. R4.1** to the reference list.)

26. Librandi, G., Tubaldi, E. & Bertoldi, K. Programming nonreciprocity and reversibility in multistable mechanical metamaterials. *Nature Commun.* **12**, 3454 (2021).

#2. In their introduction, the authors claim that “advanced processing of mechanical information in existing designs is hindered by its dissipation when interaction with networked logic gates.” This is an interesting claim, however a methodical exploration within the paper is not clear. Instead, the authors identify their system as “seamless” and “integrated” which are important attributes of an actualized system, but do not strongly indicate novel scientific work.

Response: Most mechanical computing systems consisting of bistable elements—which particularly rely on mechanical binary information—have bistable properties, characterized by two characteristic potential energies. Hence, the simplest way of describing signal transmission and/or computation based on mechanical solitary waves, driven by the cascaded non-reciprocity of soft bistable unit elements, probably appertains to the relations of two characteristic energy levels between closely coupled mechanical functional components, such as mechanical transmission lines and mechanologics (see **Fig. R4.1**). In our manuscript, these two are named as (i) the input energy barrier (U_b), the minimum energy required to achieve signal propagation (or information processing/computing) through the functional component, and (ii) the output transmissive energy (U_{out}), the energy transferred to the output end of the component (see **Fig. 2** and **Fig. R4.1a,b**). In this regard, we found that any mechanical computation event can be simplified and physically implemented only if we could calculate, design, and fabricate the computing system with engineered U_b and U_{out} for every functional component. Dissipation (or loss) of mechanical information in such a system occurs when the propagating signal with U_{out} encounters any mechanical component X (or device) having U_b^X larger than U_{out} . In this situation, the information processing is stalled, eventually rendering the entire computation malfunctioned. As a special case, also shown in **Fig. 2b** and **Fig. R4.1d**, U_{out} remains constant on average within the mechanical transmission line of sufficient length—thus, named U_{out}^T —where mechanical signals propagate at a constant speed; this is because energy dissipation due to material viscoelasticity and friction is compensated by non-reciprocal energy transmission between the bistable states of each beam (higher to lower state). Extending this logistics landscape to advanced processing of mechanical information, so-called ‘integrated mechanical computing’, we see that mechanologics with structural heterogeneity and complexity are generally prone to having U_b larger than U_{out}^T , posing the difficulty of successful computation with mechanical transition waves (**Fig. R4.2a-(iii),b**); and that their spatial assembly and network, which are necessary to attain advanced computing functions, further complicates the problem (**Fig. R4.2a-(iv),b**). No study to date has attempted to demonstrate methodical exploration on this problem, though of importance. Difficulties in quantitative analysis, design and fabrication made this even seem like an intractable problem.

As the reviewer well pointed out in the above comment #1, our study achieves an important step towards computational anatomy of intelligent soft matter by solving the issue of solitary wave-based computational propagation through 1D/2D networks of mechanologics. From a perspective of characteristic potential energy, we have methodically explored, reasoned, and developed the criterion for the propagation of mechanical signals through networked mechanologics to support our results. Given the rigor, effectiveness and experimental feasibility of our approach, we strongly believe that our achievement does indicate a novel scientific work. However, we still agree with the reviewer that the *Abstract*, *Introduction*, and

some of *Results* sections of the original manuscript were focused in part on the less important terms like “seamless” and “integrated”, and that the description of our methodical exploration was not clear enough. We have taken this opportunity to clarify the detail on our methodical exploration as follows:

- 1) By introducing two characteristic energy terms (U_b and U_{out} , as described above), we simplify the way of analyzing the behavior of solitary wave-based mechanical computing processes (**Figs. R4.1** and **R4.2**).
- 2) We show that the soliton width ($W_{soliton}$) governs the characteristics of the mechanical transmission line (N bistable beams) (see **Fig. 2a,b**). In other words, the characteristics of propagating mechanical signals in the form of solitary waves through the transmission line are the same when its length (aN , where a is the length of the unit square lattice) is longer than $W_{soliton}$ or equivalently, $N > N_{soliton}$. Based on this finding, we obtain the quantitative values of characteristic potential energies of the transmission line, $(U_b^T, U_{out}^T) \approx (U_b^{T(N_{soliton})}, U_{out}^{T(N_{soliton})}) = (0.47 \text{ mJ}, 0.73 \text{ mJ})$. Please see **Fig. 2a-e** and **Supplementary Figs. 5 and 6** in the original manuscript.
- 3) We show that the soliton width ($W_{soliton}$) governs the physical boundary of computing processes (**Fig. 2f-m**). In other words, when the propagating signal with $N_{soliton}$ interacts with any structurally heterogeneous functional component X (e.g., mechanologics, redirector units, bifurcation units, etc.), the physical boundary (or length) of the computing process becomes $\sim W_{soliton}$ ($=aN_{soliton}$) only if the effective length of X is sufficiently shorter than $W_{soliton}$ (see **Fig. 2f-m**, **Supplementary Figs. 8 and 9**). In addition, by extending this claim, we verify that the sufficient condition for dissipation-free computational propagation through the mechanologic X can be simplified as $U_{out}^{T(N_{soliton})} > U_b^{X+T} \approx U_b^{X+T(N_{soliton})}$. Please see **Fig. 2g** and **Fig. 3d**.
- 4) We propose and successfully implement architected designs and functions of mechanologics and other mechanical components that can be compactly formulated within a unit square (a), thereby satisfying the prerequisite, $a \ll W_{soliton}$ ($=aN_{soliton}$) as stated in the (3). Please see **Fig. 3** and **Supplementary Figs. 9 and 10**, and **Fig. R4.2a-(ii), b**.
- 5) Based on our findings (3) and (4), we show that the coupled energy barrier problem stemming from a network of mechanologics can be decoupled into individual computing steps. As a consequence of this stratagem, the problem of cascaded computation becomes equivalent to its minor version, as described in the finding 3). Please see **Fig. 4** and **Fig. R4.2a-(iv, v), b**.
- 6) Lastly, we experimentally demonstrate the effectiveness of the above methodical exploration for cascaded computational function, and showcase a prototype of intelligent, autonomous soft machines.

Taking the above discussion and summary of our methodical exploration into account, we have rewritten our *Abstract* and *Introduction* to more accurately summarize our findings and observations, as well as strengthened the explanation of our methodical exploration for networked logic gates in the *Results* section.

Fig. R4.1 | Two characteristic energy terms utilized for our quantitative and methodical exploration. **a**, Schematic of a unit bistable element with programmable characteristic energy components: the input energy barrier, U_b^1 , and the output transmissive energy, U_{out}^1 . **b**, Programmable non-reciprocity of the bistable beam in terms of U_b^1 and U_{out}^1 . In our system, bistable beams are designed with characteristic energies, $U_b^1 < U_{out}^1$, in order to achieve autonomous solitary wave transmission. **c-e**, Schematics and the corresponding characteristic energy components of two beams connected by a spring (**c**), a transmission line consisting of N beams (**d**), and a transmission line with the insertion of a computing unit X (**e**).

Fig. R4.2 | Overview of the challenge in cascaded mechanical computing and our solution. **a**, A set of schematics describing solitary wave-based mechanical computing processes: solitary information that propagates through mechanical transmission lines with (i) the input energy

barrier, U_b^T , and the output transmissive energy, U_{out}^T , satisfying $U_b^T < U_{out}^T$, (ii) a mechanologic having small $U_b^{X_s}$ such that $U_{out}^T > U_b^{X_s+T}$, which permits computational propagation, (iii) a mechanologic having large $U_b^{X_l}$ such that $U_{out}^T < U_b^{X_l+T}$, which leads to the transmission failure, that is, information dissipation, (iv) a cascade of mechanologies characterized by accumulated mechanical impedance that increases the input energy barrier for computation, and (v) our design approach featuring a cascade of rationally designed mechanologies to alleviate the coupled U_b issue. **b**, Qualitative U_b profiles as a function of position for the solitary wave propagations shown in **a**.

Modification to the manuscript:

(We revised the *Abstract* as follows.)

(Original abstract)

Intelligent systems built upon conventional electrical architectures often face inherent hurdles for interfacing with non-electrical signals, which are common in biological organisms and soft machines. With the recent emergence of mechanical computing, completely mechanical logic computation within architected soft materials offers a new modality to formulate computational autonomy in intelligent matter without any external powering or active elements. However, advanced processing of mechanical information in existing designs is hindered by its dissipation when interacting with networked logic gates. Here, we present a design strategy for computational soft metamaterials that provide a seamless and integrated mechanical computing interface for autonomous soft machines. Based on the systematic design and assembly of soft bistable elements, we explore how nondispersive mechanical solitary waves can compute cascaded logical functions by propagating through a network of structurally heterogeneous mechanical computing units. The developed integrated mechanical computing systems are shown to receive, transmit and compute mechanical stimuli to actuate intelligent soft machine prototypes in a seamless and integrated manner. These findings would pave the way for future intelligent robots and machines that perform computational operations between various non-electrical environmental inputs.

⇒ (Revised form)

Mechanical computing offers a new modality to formulate computational autonomy in intelligent matter **or machines** without any external powering or active elements. **Transition (or solitary) waves, induced by nonreciprocity in mechanical metamaterials comprising a chain of bistable elements, have proven to be a key ingredient for dissipation-free transmission and computation of mechanical information.** However, advanced processing of mechanical information in existing designs is hindered by its dissipation when interacting with networked logic gates. Here, **we present a metamaterial design strategy that allows non-dispersive mechanical solitary waves to compute multi-level cascaded logic functions, termed ‘integrated mechanical computing’, by propagating through a network of structurally heterogeneous computing units. From a perspective of characteristic potential energy, we establish an analytical framework that helps in understanding the solitary wave-based mechanical computation, and governs the mechanical design of key determinants for realizing cascaded logic computation, such as soliton profile and logic elements.** The developed integrated mechanical computing systems are shown to receive, transmit and compute mechanical **information** to actuate intelligent soft machine prototypes in a seamless and integrated manner. These findings would pave the way for future intelligent robots and machines that perform computational operations between various non-electrical environmental inputs.

(In page 4, 2nd paragraph: we revised the paragraph as follows.)

Here, we present a design framework for computational mechanical metamaterials that enable multi-level cascaded logic computation of propagating mechanical signals. Taking inspiration from a mature design paradigm of electronic system architectures (Fig. 1a), we demonstrate how the systematic assembly of soft bistable unit elements leads to a monolithic, integrated computing architecture consisting of a network of mechanologics connected via transmission lines (Fig. 1b). In this architecture, digital information is transmitted in the form of non-dispersive mechanical solitary waves, and its computation is performed by the dynamic mechanical coupling of the solitary waves with structural heterogeneity in mechanical computing units. Two characteristic energy terms, the input energy barrier (U_b) and the output transmissive energy (U_{out}), are introduced to develop a simplified model of solitary wave-based mechanical computation (Fig. 1c). Based on the model, we show that the soliton width, $W_{soliton}$, is the critical factor governing the behavior of both signal transmission and computation, and explore how to design transmission lines and mechanologics through which mechanical information performs dissipation-free cascaded computing without the use of any active elements (Fig. 1d,e). These findings allow us to generalize a design rule for implementing system-level integrated mechanical computing. Proof-of-concept demonstrations show how the developed mechanical computing system can direct mechanical instructions across environmental inputs, computational operations and actuator modules in a monolithic form.

(In page 4, 3rd paragraph: we revised the *Results* subsection “Design and characterization of mechanical transmission lines” as follows.)

Design and characterization of mechanical transmission lines

Key ingredients to build integrated computing architectures generally represent mechanical transmission lines and logic devices, termed “mechanologics” (Fig. 1b and Supplementary Fig. 1). On top of the classical bistable beam theory⁴¹⁻⁴³ and the mechanical solitary wave theory^{11,35}, first, we developed a design and characterization strategy of the mechanical transmission line to better understand its quantitative characteristics in a methodical way (Fig. 2). As a unit constituent, a soft bistable element with non-reciprocity was defined by two symmetric tilted beams that are connected at the center, and its possible design space was investigated by both numerical and experimental parametric studies about length (L), beam thickness (t), and angle (θ) (Extended Data Fig. 1). Note that unlike most previous studies using tilted beams^{10,11,13}, in our approach, the lattice constant (a) should additionally be considered to properly engineer the bistable characteristics⁴⁴ (Extended Data Fig. 1 and Supplementary Fig. 2). Given the experimental feasibility and geometrical stability, the optimized beam geometry was chosen to have a parameter set of $(a, L, t, \theta) = (9 \text{ mm}, 12 \text{ mm}, 0.6 \text{ mm}, 25^\circ)$ featuring a net transmissive energy $\Delta U^1 (=U_{out}^1 - U_b^1)$ of $\sim 0.15 \text{ mJ}$ (Fig. 1c, Extended Data Fig. 1 and Methods for optimal design consideration). A 1-dimensional (1D) series connection of the optimized bistable elements via spring-like linear coupling units constructed a mechanical transmission line which can produce a non-dispersive mechanical solitary wave by a chain of non-reciprocal snap-through dynamics¹¹ (Fig. 2a; see also Fig. 1d-(i), Extended Data Fig. 2 and Supplementary Video 1). To quantify the nature of this solitary wave propagation, we established a simplified model of the transmission line by introducing two characteristic energy components, U_b and U_{out} (Fig. 2b; see also Fig. 1d-(i) and Methods). Specifically, given the transmission line consisting of $N (=N_1+N_2)$ bistable elements, we reasoned that both the input energy barrier, $U_b^{T(N)}$, and the transmissive energy delivered to the output end of the transmission line, $U_{out}^{T(N)}$, are heavily reliant on the soliton width ($W_{soliton}$)⁴⁵ or equivalently the parameter $N_{soliton} (\approx W_{soliton}/a)$, the number of springs constituting a soliton at any given time frame. This implies that $U_b^{T(N)}$ and $U_{out}^{T(N)}$ can

be approximated respectively as $U_b^{T(N_{soliton})}$ and $U_{out}^{T(N_{soliton})}$, suggesting the sufficient condition for the dissipation-free transmission to be $U_{out}^{T(N_{soliton})} > U_b^{T(N_{soliton})}$ (Fig. 2b; see Methods for the details). The experimental characterization of transmission lines with varying N reveals that, with the increase of N , both $U_b^{T(N)}$ and $U_{out}^{T(N)}$ asymptotically reach steady-state values, validating the existence of $N_{soliton}$ (Fig. 2c,d and Extended Data Fig. 3). We also see that the $N_{soliton}$ value and the absolute potential energies significantly change as a function of spring stiffness, k (Fig. 2c,d and Extended Data Fig. 3). This trend is in good agreement with numerical results showing that the $N_{soliton}$ value can be engineered from 2 to 5 in average (temporarily up to 7) as a function of the normalized spring stiffness (k/k_0 , with k_0 equal to 180 N m^{-1}) (Fig. 2e and Extended Data Fig. 4). With the fixed beam geometry, therefore, the spring stiffness (k) becomes the only decisive factor governing the behavior of the transition wave-based signal transmission²⁵ (see Supplementary Fig. S3 for the effects of k and linkage geometry on solitary wave motion). **Based on the above investigations, small k ($=180 \text{ N m}^{-1}$) was adopted for our transmission line design**, which provides not only a relatively low energy barrier ($\approx 0.47 \text{ mJ}$) for propagation, but also small $N_{soliton}$ ($=2-3$) for compact system design. The resultant specification of the transmission line features $N_{soliton}$ of 2-3, U_b^T of $\sim 0.47 \text{ mJ}$ and U_{out}^T of $\sim 0.73 \text{ mJ}$, satisfying the sufficient condition for propagation $U_{out}^T > U_b^T$.

(In page 6, 2nd paragraph: we revised the *Results* subsection “Computational propagation through a structurally heterogeneous computing unit” as follows.)

A transition wave-based computing event, namely “computational propagation”, occurs when the propagating soliton meets the structural heterogeneity of logic elements (Fig. 2f and Supplementary Video 1). The presence of such heterogeneity locally increases the effective mechanical impedance (R) compared to that of the transmission line unit (R_0) (Supplementary Fig. 8), and accordingly induces transient changes in soliton profile as a consequence of the computing process (Fig. 1c). Note that R is closely correlated with U_b , but represent the effect of each segment unit more intuitively. To systematically investigate this computational propagation, we modified the model shown in Fig. 2b by inserting a mechanologic component X (Fig. 2g), and carried out numerical simulations in which the logic X was abstracted as a spring with variable mechanical impedances ($R > R_0$) (Supplementary Fig. 8 and Supplementary Video 2). Figure 2h-m shows typical numerical results involving the evolution of the normalized displacement for each bistable unit in computational propagations with R ranging from R_0 to $20R_0$. Herein, the normalized displacement is defined by the net displacement of each bistable beam (i.e., $|x_i - x_{i0}|$, where x_i is the position of the i^{th} bistable element with its initial position x_{i0}) in the propagation direction divided by the average maximum displacement of all the actuated beams (d_{max}). To visualize the shape and time-dependent propagation of solitary signals, we used the instability factor (S_i)¹¹, which implies the beam’s normalized distance from the nearest stable configuration, defined by $\min(|(x_i - x_{i0})/d_{max}|, |(x_{i0} + d_{max} - x_i)/d_{max}|)$. Notably, the obtained S_i profiles suggest that although time-dependent S_i was altered with respect to the homogeneous case (Fig. 2h,i), the soliton width ($W_{soliton}$ or $N_{soliton}$) remained nearly unchanged for all computing events in every time domain of interest (Fig. 2j-m and Supplementary Fig. 8l). This result verifies that $W_{soliton}$ prescribes the physical boundary of solitary wave-based mechanical computation as well as transmission. Our finding further suggests that if the length of the mechanologic X is sufficiently shorter than $W_{soliton}$, the sufficient condition for successful computational propagation can be simplified as $U_{out}^{T(N_{soliton})} > U_b^{X+T(N_2)} \approx U_b^{X+T(N_{soliton})}$ (Fig. 2g and Supplementary Fig. 9).

(In page 7, 1st paragraph: we revised the *Results* subsection “Design and characterization of mechanologics” as follows.)

To meet the above criterion for dissipation-free computational propagation, we devised a rational design strategy for mechanologics. Differentiated from other previous studies^{11,12,15,27,28}, we focused on developing the most compact design of force transmission by which at most four bistable beams achieve various sequences of buckling mode transitions within a unit square lattice (length, a) (Fig. 3a). Based on this backbone structure, subtle tuning of each of their potential energy profiles led to a complete set of logic functions (NOT, AND, and OR) within the unit dimension that exactly matches the transmission line unit (Fig. 3a,b, Extended Data Fig. 6, Supplementary Fig. 4 and Methods). The benefit of this mechanologic design allows for (i) superior system compactness, (ii) size matching and seamless connection with transmission lines, (iii) systematic analysis of computational propagations, and (iv) simplification of energy barrier design for solitary wave-based mechanical computing (Supplementary Fig. 5). In the resulting design, the mechanical solitary wave transmitted through the transmission line is seamlessly coupled with the mechanologics’ beams, and subsequently creates computational functions as a consequence of combinatorial buckling behaviors (Fig. 3b and Supplementary Videos 3-5). The feasibility of computational propagation was experimentally analyzed by the input/output characteristics of the mechanologics with transmission lines ($N_2=6$) connected after them (Fig. 3c). All logic devices were carefully designed to have bistable characteristics with input/output potential energies satisfying the criterion for propagation: their energy barriers were engineered to be lower than U_{out}^T (≈ 0.73 mJ), thereby permitting successful computational propagation without loss of information (Fig. 3d). Importantly, we note that the experimentally obtained values of energy barriers ($U_b^{X+T(N_2)}$) start to saturate at the regime of $N_2=N_{soliton}$ ($=2-3$) (Fig. 3d), which perfectly matches our finding that $N_{soliton}$ is a key decisive factor for transition wave-based computational propagations. The operating performance of the proposed mechanologics shows explicit computational propagations which were quantitatively analyzed by the evolution of normalized displacement of each bistable beam (Fig. 3e, Supplementary Fig. 6 and Supplementary Videos 3-5).

(From page 7, 2nd paragraph to page 8, 2nd paragraph: we revised the *Results* subsection “Computational propagation through multiple cascaded mechanologics” as follows.)

Achieving integrated computing is equivalent to addressing the effective impedance accumulated from multiple cascaded logic devices (Fig. 1c,d). In general, the first energy barrier of the 1D cascaded computational system (U_b^{1st}) can be expressed by (Fig. 4a):

$$U_b^{1st} = U_b^{\sum\{X_i+T(L_i)\}} \approx U_b^{\sum\{X_i+T(N_{cas,i})\}} \quad (1)$$

where X_i is the i^{th} mechanologic, L_i the length of the transmission line that bridges the adjacent computing units X_i and X_{i+1} , and $N_{cas,i}$ the number of intermediate cascaded springs, calculated as $[L_i/a]$ with the bracket indicating the rounding function to an integer. Despite this cumulative form, our finding on the rule for computational propagation, as shown in Fig. 2g, suggests that such coupled energy barrier problems can be decoupled into individual computing steps only if the dimension of mechanologics is smaller than the soliton width ($\approx aN_{soliton}$) and $N_{cas} \geq N_{soliton}$ holds, which reads (see also Fig. 1c,d):

$$U_b^{k^{th}} = U_b^{\sum_{i=k}^{\infty}\{X_i+T(N_{cas,i})\}} \approx U_b^{X_k+T(N_{soliton})}. \quad (2)$$

To quantitatively validate this approach, we conducted a set of numerical studies investigating the behavior of computational propagation through multiple cascaded mechanologics with various

effective impedances and N_{cas} (Fig. 4b). The results suggest that while **cascaded systems with small N_{cas} ($< N_{soliton}$) failed in transmission as well as computation through the logics** with relatively large R due to their coupled energy barrier effects (Fig. 4c-(i, ii), and Supplementary Video 6), the increase in N_{cas} successfully settled down the barrier, enabling the signal to smoothly pass through the **logics** with larger R (Fig. 4c-(iii, iv), and Supplementary Video 6). (...)

(In page 10, 2nd paragraph: we revised the paragraph as follows.)

We have developed a design framework for metamaterial architectures that could execute multi-level cascaded computational functions of mechanical information using purely mechanical computing processes. Our approach of modeling and analyzing computational propagations of solitary waves, together with the compact, rational and seamless design of mechanologies, allowed us to overcome major challenges in cascaded computing where solitary information can be transmitted through a network of mechanologies in a lossless manner. The outcome was the electronics-free, integrated mechanical computing that realized integrative mechanical-to-mechanical operations directly mapping environmental mechanical stimulus inputs to soft actuators.

(Parts of **Figs. R4.1 and R4.2** were added to **Figure 1**; and the caption was accordingly revised as follows.)

Fig. 1 | Design concept of integrated mechanical computing. **a**, Unit cells, their topological assembly for computational logics, and system architectures for integrated mechanical computing. **b**, Schematic of a unit bistable element with programmable characteristic energy components: the input energy barrier, U_b^I , and the output transmissive energy, U_{out}^I . **c**, A set of schematics describing solitary wave-based mechanical computing processes: solitary information that propagates through mechanical transmission lines with (i) the input energy barrier, U_b^T , and the output transmissive energy, U_{out}^T , satisfying $U_b^T < U_{out}^T$, (ii) a mechanologic having small U_b^{Xs} such that $U_{out}^T > U_b^{Xs+T}$, which permits computational propagation, (iii) a mechanologic having large U_b^{Xl} such that $U_{out}^T < U_b^{Xl+T}$, which leads to the transmission failure, that is, information dissipation, (iv) a cascade of mechanologies characterized by accumulated mechanical impedance that increases the input energy barrier for computation, and (v) our design approach featuring a cascade of rationally designed mechanologies to alleviate the coupled U_b issue. **d**, Qualitative U_b profiles as a function of position for the solitary wave propagations shown in **c**.

(Figure 2 and its caption were revised as follows.)

Fig. 2 | Design and characterization of solitary wave-based computational propagation. **a**, Typical **propagation** behavior of mechanical **solitary information** through a **mechanical** transmission line (Supplementary Video 1). Scale bar, 1 cm. **b**, Schematic diagram of the mechanical transmission line consisting of N ($=N_1 + N_2$) elements with input, U_b^T , and output, U_{out}^T characteristic energy components (see Methods). **c,d**, Experimentally obtained U_b^T and U_{out}^T of transmission lines with N varying from 0 to 9. **e**, Numerically estimated $N_{soliton}$ as a function of the normalized spring stiffness (k/k_0) with $k_0=180 \text{ N m}^{-1}$. **f**, Typical **propagation behavior** of mechanical **solitary information** through a transmission line that contains a **structurally heterogeneous computing unit** (Supplementary Video 1). Scale bar, 1 cm. **g**, Schematic diagram of the transmission line connected with a mechanical computing unit X (see Methods). **h,j,l**, Numerical studies on the evolution of the normalized displacement of each bistable unit in homogeneous transmission (mechanical impedance, R_0) (**h**) and computational transmission through a computing unit having a mechanical impedance (R) of $10R_0$ (**j**) and $20R_0$ (**l**) (Supplementary Video 2). The variable mechanical impedance is defined only for the tenth spring. A color bar indicates normalized displacement. **i,k,m**, The time-evolving propagation behavior of the instability factor (S_i) along with the system shown in **h,j,l**, respectively. All error bounds indicate SD.

(Figure 4 and its caption were revised as follows.)

Fig. 4 | Computational propagation through one-dimensionally (1D) cascaded mechanologics. **a**, Schematic description of the general design rule for cascaded computation. **b**, Simulation results of cascaded computations through a 1D network of mechanologics (mechanical impedance, R) connected by transmission lines (mechanical impedance, R_0) with varying N_{cas} (0 to 5). The normalized time taken to complete the propagation was introduced not only to characterize the propagation performance, but to indirectly estimate the effective mechanical impedance of overall cascaded systems. **c**, Numerical studies on the evolution of the normalized displacement of each bistable unit in computational propagations through cascaded mechanologics (mechanical impedance, $R=15R_0$) with $N_{cas}=0$ (i), 1 (ii), 3 (iii), 5 (iv) (Supplementary Video 6). **d**, Experimental verification of the proposed design rule for cascaded computing (Supplementary Video 7). To demonstrate 1D cascaded computing, mechanical inverters with two different phases ($\phi=0, \pi$) were connected in series. The time parameter, which has a linear relationship with mechanical impedance for propagation, was obtained for every bistable beam to characterize the behavior of cascaded computing. Scale bar, 2cm.

(The original Fig. 1a was moved to Supplementary Fig. 1; and the caption was accordingly revised as follows.)

Supplementary Fig. 1 | Inspiration and design requirements for integrated mechanical computing. **a**, Unit cells, their topological assembly for computational logics, and integrated system architectures for conventional electrical computing **b,c**, A representative mechanical analogue of integrated computing (**b**) and its schematic diagram (**c**). As in its electrical counterpart, the integrated system architecture for mechanical computing needs to combine networked mechanical computing units that map environmental input mechanical stimuli to the output mechanical interactions. **d**, Key design requirements for integrated mechanical computing are threefold: (i) The optimal mechanical transmission line needs to be developed for stable propagation of mechanical information. The engineering of its input/output characteristics is also critical to deal with output load effects induced by the effective mechanical impedance of a single computing unit or its cascaded form. (ii) Rational designs of mechanical computing units (“mechanologics”), which allow for not only compact, deterministic computational functions, but also accessible modular connectivity, need to be developed. (iii) A systematic approach for achieving lossless computation through multiple cascaded mechanologics needs to be established.

#3. The detailed description of the biological example from which the authors drew inspiration distracted from the central claim of the paper: demonstrating multi-layered signal propagation. The authors could pare back the biological discussion and instead add detail to other sections.

Response: We fully agree with the reviewer’s suggestion. According to the reviewer’s comment, we have pared back the biological discussion and instead added details of multi-layered signal propagation in the *Results* section.

Modification to the manuscript:

(From page 9, 3rd paragraph to page 10, 2nd paragraph: we revised the subsection of *Results* as follows.)

Bioinspired autonomous soft machines based on integrated mechanical computing

The proposed design principle for integrated mechanical computing provides an effective computing paradigm that physically instantiates intricate computational abstractions between mechanical inputs and outputs. To show potential soft machines interfaced by such a computational framework, we present a proof-of-principle demonstration of Mimosa plant (*M. pudica*)-inspired autonomous, intelligent, integrative soft machines (Fig. 6). A close observation of *M. pudica*’s thigmonastic movements shows that the plant’s response to the mechanical input is differentiated depending on the intensity of the stimulus^{6,7,46,47} (Fig. 6a,b and Supplementary Video 9). We reinterpreted this stimuli-responsive operating mechanism by means of integrated mechanical computing to realize **electronics-free** intelligent soft machines (Supplementary Fig. 15).

First, we developed a soft actuator **consisting** of a 3D printed base layer and an active hydrogel layer, with a design that maximized the actuation rate and strain upon water absorption (Fig. 6c and Methods). The monolithically fabricated reservoir could transport a water droplet (~30 μ L) to the actuator (Fig. 6d and Supplementary Fig. 16), which in turn leads to *M. pudica*-like shape morphing with a bending angle of up to 50° (Fig. 6e, Supplementary Fig. 17 and Supplementary Video 10). A series connection of ten soft actuators via a transmission line demonstrated a soft machine, where a single mechanical input triggered the sequential propagation of actuation events, reminiscent of one branch of *M. pudica* (Fig. 6f and Supplementary Video 11). **Additionally, we implemented an**

integrated mechanical computing circuit whose logic function was based on the truth table that maps mechanical outputs to three inputs (S1, S2, S3) with different energy barriers (Supplementary Fig. 18a-d). The developed integrative machine consists of 17 soft actuators (arranged within two different branches) and 104 bistable elements in total whose strategic assembly constructs a computational network between 4 mechanologies and 13 redirector components (Fig. 6g and Supplementary Fig. 18e). **This indicates that an initial environmental trigger can demonstrate computational propagations of mechanical information through a network of 17 structurally heterogeneous mechanical units including 4 mechanologies.** Note that the whole circuit was designed in a double-layer layout using two sub-level circuits (Supplementary Fig. 18); therefore, mechanical signals can propagate through the two layers independently and without interacting with each other, enabling us to reduce the areal footprint as well as allow for multi-channel signal crossings. With input stimuli S2 and S3, the signal initiates concurrent and independent propagations along two different layers to actuate the two parts of the branch 1 simultaneously, whereas for the input S1 the signal propagates along only one layer and actuates only one part of the branch 1 (Fig. 6h,i and Supplementary Video 12). **This intelligent** operation of the machine shows that depending on the input type, proper mechanical instructions could be propagated through and interact seamlessly with computational operations and actuator modules to demonstrate the force-dependent actuation sequences observed in *M. pudica*.

(We revised the subsection of *Methods* as follows.)

Fabrication of ~~Mimosa-inspired~~ soft hydrogel actuators

A solution of photoinitiator was prepared by dissolving 150 mg of 2,2-dimethoxy-2-phenylacetophenone (99%, Sigma Aldrich) in 1 mL of dimethyl sulfoxide (Sigma Aldrich). Ethylene glycol dimethacrylate (100 μ L) and the photoinitiator solution (150 μ L) were added in 2 ml of poly(ethylene glycol) diacrylate (PEGDA, Mn 575, Sigma Aldrich). An 8 μ L of the prepared PEGDA solution was dropped onto the surface of 3D printed actuator body made up of the Elastic 50A material (25 mm \times 4 mm) and then covered with a slide glass to adjust its thickness to 25 ± 5 μ m. The confined solution was polymerized by irradiation of UV light (Bio-Link 365, wavelength of 365 nm, Vilber Lourmat GmbH) for 30 min.

(We revised the caption of **Fig. 6** as follows.)

Fig. 6 | ~~Bioinspired~~ Autonomous soft machines based on integrated mechanical computing. a, The force-dependent thigmonastic movements of *M. pudica* (Supplementary Video 9). **b,** Schematic illustration of the detailed interpretation of *M. pudica*'s intelligent behavior shown in **a**. **c,** Schematic illustration of the design and actuation mechanism of ~~Mimosa-inspired~~ soft hydrogel actuators. **d,** Photographs of the reservoirs mounted monolithically on the transmission line for water droplet transport with a volume up to 30 μ L. **e,** Typical actuation behaviors of the hydrogel actuator upon water uptake with volume of 10, 20, and 30 μ L, respectively from left to right (Supplementary Video 10). **f,** A ~~Mimosa-inspired~~ soft machine consisting of ten soft actuators connected in series via the mechanical transmission line (left) and its sequential actuation upon mechanical stimulation (right) (Supplementary Video 11). **g,** Photograph of the ~~bioinspired~~ autonomous soft machine that reproduces the function of the internal operating circuit of *M. pudica* as described in **b**. The system consists of double-layer integrated mechanical computing circuits (1 AND gate and 3 OR gates) and 17 soft actuators arranged within two different branches. **h,i,** Operation of the integrated mechano-intelligence (**h**) and resulting force-dependent actuation sequences (**i**) (Supplementary Video 12). Scale bars, 1 cm (**d-f**) and 2 cm (**g**).

REVIEWERS' COMMENTS

Reviewer #4 (Remarks to the Author):

The authors thoroughly responded to the main comments and their edits improved the quality of the paper. In particular, changes to figure 1 and the addition of the energy barrier and position plot as well as by detailing the two characteristic energy terms improved the rigor of analysis to back up claims made in the abstract and introduction.